# Behavioural and neural signatures of perceptual decision-making are modulated by pupil-linked arousal

Jochem van Kempen[1,2]*, Gerard M Loughnane[3,4], Daniel P Newman[2], Simon P Kelly[5], Alexander Thiele[1], Redmond G O'Connell[2,4,6], Mark A Bellgrove[2,6]

[1]Institute of Neuroscience, Newcastle University, Newcastle upon Tyne, United Kingdom; [2]Monash Institute for Cognitive and Clinical Neurosciences, School of Psychological Sciences, Monash University, Melbourne, Australia; [3]School of Engineering, Trinity College Dublin, Dublin, Ireland; [4]Trinity College Institute of Neuroscience, Trinity College Dublin, Dublin, Ireland; [5]School of Electrical and Electronic Engineering, University College Dublin, Dublin, Ireland; [6]School of Psychology, Trinity College Dublin, Dublin, Ireland

**Abstract** The timing and accuracy of perceptual decision-making is exquisitely sensitive to fluctuations in arousal. Although extensive research has highlighted the role of various neural processing stages in forming decisions, our understanding of how arousal impacts these processes remains limited. Here we isolated electrophysiological signatures of decision-making alongside signals reflecting target selection, attentional engagement and motor output and examined their modulation as a function of tonic and phasic arousal, indexed by baseline and task-evoked pupil diameter, respectively. Reaction times were shorter on trials with lower tonic, and higher phasic arousal. Additionally, these two pupil measures were predictive of a unique set of EEG signatures that together represent multiple information processing steps of decision-making. Finally, behavioural variability associated with fluctuations in tonic and phasic arousal, indicative of neuromodulators acting on multiple timescales, was mediated by its effects on the EEG markers of attentional engagement, sensory processing and the variability in decision processing.
DOI: https://doi.org/10.7554/eLife.42541.001

*For correspondence:
jochem.van-kempen@ncl.ac.uk

Competing interests: The authors declare that no competing interests exist.

## Introduction

The speed and accuracy with which humans, as well as non-human animals, respond to a stimulus depends not only on the characteristics of the stimulus, but also on the cognitive state of the subject. When drowsy, a subject will respond more slowly to the same stimulus compared to when she is attentive and alert. Central arousal also fluctuates across a smaller range during quiet wakefulness, when the subject is neither drowsy or inattentive, nor overly excited or distractible. Although these trial-to-trial fluctuations can impact on behavioural performance during decision-making tasks (*Aston-Jones and Cohen, 2005*), it is largely unknown how arousal modulates the underlying processes that support decision formation. Perceptual decision-making depends on multiple neural processing stages that represent and select sensory information, those that process and accumulate sensory evidence, and those that prepare and execute motor commands. Variability in central arousal could affect any one or potentially all of these processing stages, which in turn could influence behavioural performance.

The neuromodulatory systems that control central arousal state, such as the noradrenergic (NA) locus coeruleus (LC) and the cholinergic basal forebrain (BF), have also been suggested to drive fluctuations in endogenous activity linked to changes in cortical (de)synchronization, that is cortical state

**eLife digest** Driving along a busy street requires you to constantly monitor the behavior of other road users. You need to be able to spot and avoid the car that suddenly changes lane, or the pedestrian who steps out in front of you. How fast you can react to such events depends in part on your brain's level of alertness, or 'arousal'. This in turn depends on chemicals within the brain called neuromodulators.

Neuromodulators are a type of neurotransmitter. But whereas other neurotransmitters enable brain cells to signal to each other, neuromodulators turn the volume of these signals up or down. The activity of brain regions that produce neuromodulators varies over time, leading to changes in brain arousal. These changes take place over different time scales. Sudden unexpected events, such as those on the busy street above, trigger sub-second changes in arousal. But arousal levels also show spontaneous fluctuations over minutes to hours. We can follow these changes in real-time by looking into a participant's eyes. This is because the brain regions that produce neuromodulators also control pupil size.

Van Kempen et al. have now combined measurements of pupil size with recordings of electrical brain activity. Healthy volunteers learned to press a button as soon as a target appeared on a screen. The larger a volunteer's pupils were before the target appeared, the more slowly the volunteer responded on that trial. Large baseline pupil size is thought to indicate a high baseline level of brain arousal. By contrast, the larger the increase in pupil size in response to the target, the faster the volunteer responded on that trial. This increase in pupil size is thought to reflect an increase in brain arousal.

The recordings of brain activity provided clues to the underlying mechanisms. In trials with large baseline pupil size – and therefore high baseline arousal – the volunteers' brains showed more variable responses to the target. But in trials with a large increase in pupil size – and a large increase in arousal – the volunteers' brains showed less variable responses, as well as stronger signals related to attention.

Neuromodulators thus act on different timescales to influence different aspects of cognitive performance, including attention and target detection. Fluctuating levels of neuromodulator activity may help explain the variability in our behavior. Monitoring pupil size is one way to gain insights into the mechanisms that bring about these changes in neuromodulator activity.

DOI: https://doi.org/10.7554/eLife.42541.002

(*Harris and Thiele, 2011*; *Lee and Dan, 2012*), and are linked to cognitive functions such as attention (*Thiele and Bellgrove, 2018*), both known to affect information processing and behavioural performance. These modulatory systems have both tonic and phasic firing patterns that are recruited on different timescales and support different functional roles (*Aston-Jones and Cohen, 2005*; *Dayan and Yu, 2006*; *Parikh et al., 2007*; *Parikh and Sarter, 2008*; *Sarter et al., 2016*). Tonic changes in neuromodulator activity occur over longer timescales that can span multiple trials, whereas fast (task-evoked) recruitment through phasic activation occurs on short enough timescales to influence neural activity and behavioural decisions within the same trial (*Aston-Jones and Cohen, 2005*; *Bouret and Sara, 2005*; *Dayan and Yu, 2006*; *Parikh et al., 2007*).

Pupil diameter correlates strongly with a variety of measurements of cortical state and behavioural arousal (*Eldar et al., 2013*; *Reimer et al., 2014*; *McGinley et al., 2015b*; *McGinley et al., 2015a*; *Vinck et al., 2015*; *Engel et al., 2016*), and can thus be considered a reliable proxy of central arousal state. Indeed, there is a strong correlation between pupil size and activity in various neuromodulatory centres that control arousal (*Aston-Jones and Cohen, 2005*; *Gilzenrat et al., 2010*; *Murphy et al., 2014a*; *Varazzani et al., 2015*; *Joshi et al., 2016*; *Reimer et al., 2016*; *de Gee et al., 2017*). Both baseline pupil diameter, reflecting tonic activity levels in neuromodulatory centres (tonic arousal), and task-evoked pupil diameter changes (phasic arousal), have been related to specific neural processing stages of perceptual decision making. Baseline pupil diameter correlates with sensory sensitivity (*McGinley et al., 2015a*; *McGinley et al., 2015b*) and is predictive of behavioural performance during elementary detection tasks (*Murphy et al., 2011*; *McGinley et al., 2015a*). Pupil diameter also changes phasically in the course of a single decision (*Beatty, 1982a*; *de Gee*

*et al., 2014*; *de Gee et al., 2017*; *Lempert et al., 2015*; *Murphy et al., 2016*; *Urai et al., 2017*), and has been related to specific elements of the decision making process, such as decision bias (*de Gee et al., 2014*; *de Gee et al., 2017*), uncertainty (*Urai et al., 2017*), and urgency (*Murphy et al., 2016*). This suggests that these neuromodulatory systems do not only dictate network states (through tonic activity changes), but that they are recruited throughout the decision making process (*Cheadle et al., 2014*; *de Gee et al., 2014*; *de Gee et al., 2017*). Although both baseline pupil diameter and the phasic pupil response have been associated with specific aspects of decision-making, the relationship between pupil-linked arousal and the electrophysiological correlates of decision-making are largely unknown.

Recently developed behavioural paradigms have made it possible to non-invasively study the individual electroencephalographic (EEG) signatures of perceptual decision-making described above (*O'Connell et al., 2012*; *Kelly and O'Connell, 2013*; *Loughnane et al., 2016*; *Loughnane et al., 2018*; *Newman et al., 2017*). In these paradigms, participants are required to continuously monitor (multiple) stimuli for subtle changes in a feature. Because stimuli are presented continuously, target onset times (and locations) are unpredictable, and sudden stimulus onsets are absent, eliminating sensory evoked deflections in the EEG traces. These characteristics allow for the investigation of the gradual development of build-to-threshold decision variables as well as signals that code for the selection of relevant information from multiple competing stimuli, a critical feature of visuospatial attentional orienting that impact evidence accumulation processes (*Loughnane et al., 2016*).

Here, we asked how arousal influences EEG signals that relate to each of the separate processing stages described above. Specifically, we tested the effects of pupil-linked arousal on pre-target preparatory parieto-occipital $\alpha$-band activity, associated with fluctuations in the allocation of attentional resources (*Kelly and O'Connell, 2013*); early target selection signals measured over contra- and ipsilateral occipital cortex, the N2c and N2i (*Loughnane et al., 2016*); perceptual evidence accumulation signals measured as the centroparietal positivity (CPP), which is a build-to-threshold decision variable demonstrated to scale with the strength of sensory evidence and predictive of reaction time (RT) (*O'Connell et al., 2012*; *Kelly and O'Connell, 2013*); and motor-preparation signals measured via contralateral $\beta$-band activity (*Donner et al., 2009*; *O'Connell et al., 2012*). Of these signals, we extracted specific characteristics such as the latency, build-up rate and amplitude, and tested whether these were affected by pupil-linked arousal. Additionally, because the variance and response reliability of the membrane potential of sensory neurons varies with pupil diameter (*Reimer et al., 2014*; *McGinley et al., 2015a*), we also investigated whether arousal affected the inter-trial phase coherence (ITPC), a measure of across trial consistency in the EEG signal, of the N2 and the CPP.

We found that both baseline pupil diameter as well as the pupil response were predictive of behavioural performance, and that this relationship was best described by non-monotonic, but not U-shaped, second-order polynomial model fits. Furthermore, we found that both tonic and phasic arousal bore a predictive relationship with the neural signals coding for baseline attentional engagement, early target selection, decision processing as well as the preparatory motor response. Although neural activity representing all these stages varied with changes in arousal, unique variability in task performance due to tonic arousal (baseline pupil diameter) could only be explained by the amplitude of target selection signals and the consistency of the CPP, reflecting decision processing. In contrast, variability due to phasic arousal (pupil response) was explained by pre-target $\alpha$-band activity as well as the consistency of the CPP.

## Results

80 subjects performed a continuous version of the random dot motion task in which they were asked to report temporally and spatially unpredictable periods of coherent motion within either of two streams of random motion (*Figure 1A*). We investigated whether the trial-to-trial fluctuations in behavioural performance and EEG signatures of perceptual decision making could, in part, be explained by trial-to-trial differences in the size of the baseline pupil diameter (reflecting tonic arousal) and the post-target pupil response (reflecting phasic arousal). We quantified this relationship by allocating data into five bins based on the size of either the baseline pupil diameter or the phasic pupil diameter response (*Figure 1B & C*). Baseline pupil diameter was computed as the average pupil diameter over the 100 ms preceding target onset. The phasic pupillary response was

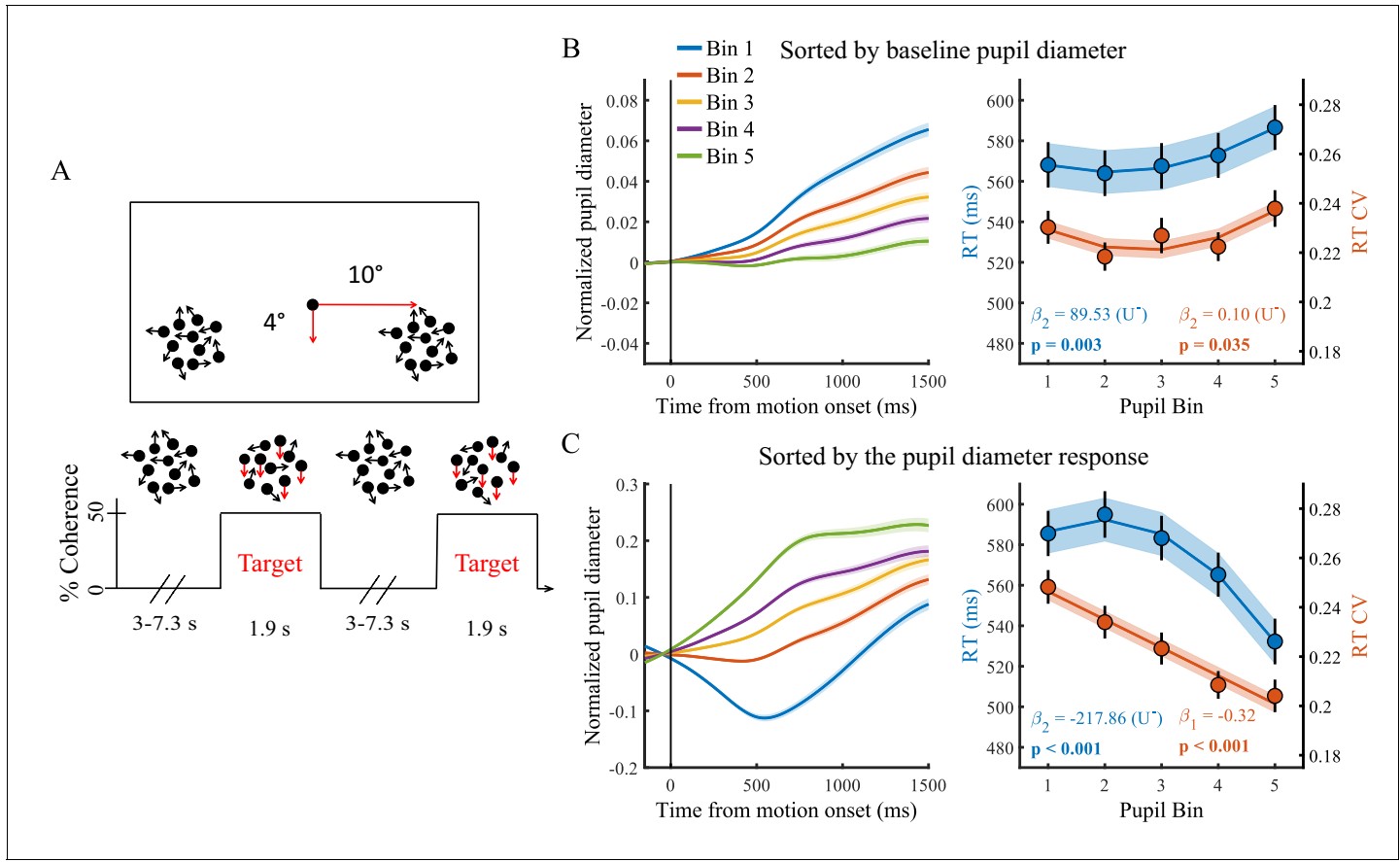

**Figure 1.** Paradigm and task performance related to baseline pupil diameter and the task-evoked pupil response. (**A**) Paradigm. Subjects fixated on a central dot while monitoring two peripheral patches of continuously presented randomly moving dots. At pseudorandom times an intermittent period of coherent downward motion (50%) occurred in either the left or the right hemifield. A speeded right handed button press was required upon detection of coherent motion. (**B**) Pupil diameter time course and task performance sorted by baseline pupil diameter. (Left) Pupil time-course for the five bins. (Right) Behavioural performance for the five bins. Markers indicate mean reaction times (RT, blue, left y-axis) and reaction time coefficient of variation (RTcv, red, right y-axis), lines and shading indicate significant model fits. (**C**) Same conventions as B, but sorted by the pupil diameter response. Error bars and shaded regions denote ±1 standard error of the mean (SEM). Stats, β weights: linear mixed effects model analyses, U: indicates presence (+) or absence (-) of significant U-shaped relationship (Statistical analyses).

DOI: https://doi.org/10.7554/eLife.42541.003

The following source data and figure supplements are available for figure 1:

**Source data 1.** Csv table containing data for *Figure 1* panel B.
DOI: https://doi.org/10.7554/eLife.42541.010
**Source data 2.** Csv table containing data for *Figure 1* panel C.
DOI: https://doi.org/10.7554/eLife.42541.011
**Figure supplement 1.** The neural input to the pupil diameter system is best described by a linear up-ramp.
DOI: https://doi.org/10.7554/eLife.42541.004
**Figure supplement 2.** Relating various measures of the phasic pupil response to behavioural performance.
DOI: https://doi.org/10.7554/eLife.42541.005
**Figure supplement 3.** Application of the linear up-ramp model on a single-trial basis.
DOI: https://doi.org/10.7554/eLife.42541.006
**Figure supplement 4.** Application of the linear up-ramp model across bins of trials.
DOI: https://doi.org/10.7554/eLife.42541.007
**Figure supplement 5.** The effect of baseline pupil diameter on the relationship between the pupil response and behavioural performance.
DOI: https://doi.org/10.7554/eLife.42541.008
**Figure supplement 6.** The relationship between baseline pupil diameter and task performance, for band-pass (0.1–6 Hz), rather than low-pass (<6 Hz) filtered pupil diameter data.
DOI: https://doi.org/10.7554/eLife.42541.009

estimated using a single trial general linear model (GLM) approach (Materials and methods). We first assessed the neural input to the peripheral pupil system by applying multiple models with onset and response components as well as various different shapes for the sustained component (*Murphy et al., 2016*) across all trials for each subject (*Figure 1—figure supplement 1*). Next we applied the grand average best-fitting model (linear up-ramp) on individual trials (*Bach et al., 2018*). This provided us with a trial-by-trial estimate of the amplitude of each temporal component. Comparison against several other measures of the pupil diameter response (*Figure 1—figure supplement 2*), controlling for variance inflation factors (*Figure 1—figure supplement 3*) and applying the same model across bins of trials, or orthogonalizing the predictors (*Figure 1—figure supplement 4*) provided support for the reliability of the estimated amplitude of the pupillary response. Here we present the relationship of the amplitude of the target onset component to the behavioural and EEG signatures of perceptual decision-making. We then used sequential multilevel model analyses and maximum likelihood ratio tests to test for fixed effects of pupil bin. We determined whether a linear fit was better than a constant fit and subsequently whether the fit of a second-order polynomial, indicating a non-monotonic relationship between pupil diameter and behaviour/EEG, was superior to a linear fit. We furthermore used a variant of the 'two-lines' approach (*Simonsohn, 2017*) to test whether any non-monotonic relationship was best described by an (inverted) U-shape.

## Both tonic and phasic arousal are predictive of task performance

We first investigated the relationship between trial-by-trial pupil dynamics and behavioural performance. As stimuli were presented well above perceptual threshold, our subjects performed at ceiling (mean, 98.7%; range: 92–100%, *Newman et al., 2017*). We therefore focused on RT and the RT coefficient of variation (RTcv), a measure of performance variability calculated by dividing the standard deviation in RT by the mean (*Bellgrove et al., 2004*), rather than accuracy. We found that baseline pupil diameter displayed a non-monotonic relationship with both measures of behavioural performance (RT $\chi^2_{(1)}$=8.84, p=0.003; RTcv $\chi^2_{(1)}$=4.43, p=0.035). Neither effects were, however, significantly U-shaped (*Figure 1B*). Rather, RT was slower on trials with higher baseline arousal levels. The pupil diameter response, on the other hand, displayed a non-monotonic (but not U-shaped) relationship with RT ($\chi^2_{(1)}$=51.89, p<0.001) and an inverse linear relationship with RTcv ($\chi^2_{(1)}$=45.94, p<0.001). For both measures, best performance was found on trials with the largest pupil responses (*Figure 1C*). This relationship remained very similar when trial-by-trial fluctuations in the pupil response that are due to variability in the amplitude or phase of the baseline pupil diameter were not removed (*Figure 1—figure supplement 5*). We furthermore repeated the sequential regression analysis in single-trial, non-binned data, in which we additionally controlled for time-on-task effects, confirming that these effects were not dependent on the binning procedure (*Supplementary file 1*). Additionally, we noticed that when we band-pass filtered the pupil diameter, rather than low-pass filtered, the relationship between baseline pupil diameter and task performance was not significant (*Figure 1—figure supplement 6*). This suggests that slow fluctuations in baseline pupil diameter (<0.01 Hz) are driving the effect on task performance.

Having established a relationship between task performance and both tonic and phasic modes of central arousal state, we next focused on the relationship between these pupil dynamics and the neural signatures underpinning target detection on this perceptual decision making task (*Loughnane et al., 2016*; *Newman et al., 2017*).

## Phasic arousal has an approximately linear relationship with decision processing

During decision making, perceptual evidence has to be accumulated over time. This accumulation process has long been related to build-to-threshold activity in single neurons in parietal cortex (*Gold and Shadlen, 2007*); but see *Latimer et al., 2015*, *Latimer et al., 2016*; *Shadlen et al., 2016*). The centro-parietal positivity (CPP) measured from scalp EEG exhibits many of these same properties, including a representation of the accumulation of sensory evidence towards a decision bound (*O'Connell et al., 2012*; *O'Connell et al., 2018*; *Kelly and O'Connell, 2013*). Because in this study we used relatively strong sensory evidence (50% coherence), it is possible that subjects may not have relied upon any temporal integration of this motion signal to reach a decision. Rather, variability in RT could be brought about by variation in the onset transient of target selection due to the

temporal and spatial uncertainty of the target stimulus. On single trials, decision formation could be a step-like signal that averaged across trials looks like an accumulate-to-bound signal (*Latimer et al., 2015*). Although we cannot discount this possibility, aligning the visual early target selection signals (N2c) to response reveals a much lower signal amplitude compared to aligning it to target onset (*Figure 2—figure supplement 1*). This indicates that there is no fixed delay between target selection and the response, and that there is variability in the duration of the sustained period of this task. This variation could indicate different trial-to-trial strategies (e.g. comparing motion in one stimulus against the stimulus on the other side of the screen), or in addition some variability in accumulation rate. Because of this uncertainty, we refer to the functional significance of the CPP as decision processing.

Here we tested the relationship between the pupil diameter response and the onset, build-up rate, amplitude and consistency (ITPC) of the CPP (*Figure 2*). We found that the onset latency of the CPP, defined as the first time point that showed a significant difference from zero for 15 consecutive time points, displayed an inverse monotonic relationship with the size of the pupil response ($\chi^2_{(1)}$=5.60, p=0.018), such that the fastest onsets were found for the largest pupil response bins (*Figure 2A*). Likewise, the build-up rate ($\chi^2_{(1)}$=4.45, p=0.035), but not the amplitude (p=0.15), of the CPP varied with the pupil response, displaying the steepest slope on trials with the largest pupil dilations. Because the membrane potential of sensory neurons shows the least variance and highest response reliability at intermediate baseline pupil diameter (*McGinley et al., 2015a*), we additionally investigated the ITPC, a measure of across trial consistency, of the CPP. We computed ITPC with a single-taper spectral analysis in a 512 ms sliding window computed at 50 ms intervals, with a frequency resolution of 1.95 Hz (Materials and methods). Based on the stimulus-locked grand average time-frequency spectrum, we selected a time (300–550 ms) and frequency window (<4 Hz) for further statistical analyses (*Figure 2C*). We found an approximately linear relationship between pupil diameter response and the consistency of the CPP signal ($\chi^2_{(1)}$=41.79, p<0.001), indicating that the CPP signal is less variable for larger pupil response bins (*Figure 2D*). This relationship was also present when we aligned the CPP to the response (*Figure 2—figure supplement 2*), indicating that this effect is unlikely to solely reflect variability in the onset of the CPP. Thus, we found that the size of the pupillary response was predictive of both the onset latency, build-up rate as well as the ITPC of the CPP. Moreover, the relationship with the neural parameters of the CPP resembled the relationship between the pupil response and behavioural performance (*Figure 1C*). Large pupil dilations were predictive of faster responses, earlier CPP onset latencies, as well as steeper build-up rates and more consistent CPP. Next, we asked whether other stages of information processing underpinning perceptual decision making also varied with the pupil response.

## The phasic pupil response relates monotonically to spectral measures of baseline attentional engagement, but not motor output

We next investigated pre-target preparatory α-band power (8–13 Hz), a sensitive index of attentional deployment that has been shown to vary with behavioural performance. Specifically, previous studies have found higher pre-target α-band power preceding trials with longer RT, and suggested that fluctuations in α-power may reflect an attentional influence on variability in task performance (*Ergenoglu et al., 2004*; *van Dijk et al., 2008*; *O'Connell et al., 2009*; *Kelly and O'Connell, 2013*). We first verified the relationship between α-band power and behavioural performance by binning the data into five bins according to α-band power and performing the same sequential regression analysis as described above (*Figure 3A*). We replicated previous findings (*Kelly and O'Connell, 2013*) and found an approximately linear relationship between α-band power and RT ($\chi^2_{(1)}$=25.27, p<0.001) but not RTcv (p=0.48). In line with previous research (*Hong et al., 2014*), pupil diameter response was inversely related to α-band power (*Figure 3B*), displaying an approximately linear relationship ($\chi^2_{(1)}$=28.24, p<0.001), suggesting that pre-target attentional engagement is related to phasic arousal.

We next focused on response-related motor activity in the form of left hemispheric β-power (LHB). LHB decreases before a button press and has been shown to reflect the motor-output stage of perceptual decision making, but also to trace decision formation, reflecting the build-up of choice selective activity (*Donner et al., 2009*). Here we investigated the LHB amplitude and build-up rate preceding response (*Figure 3C*). We found that neither LHB amplitude (p=0.63) nor LHB slope

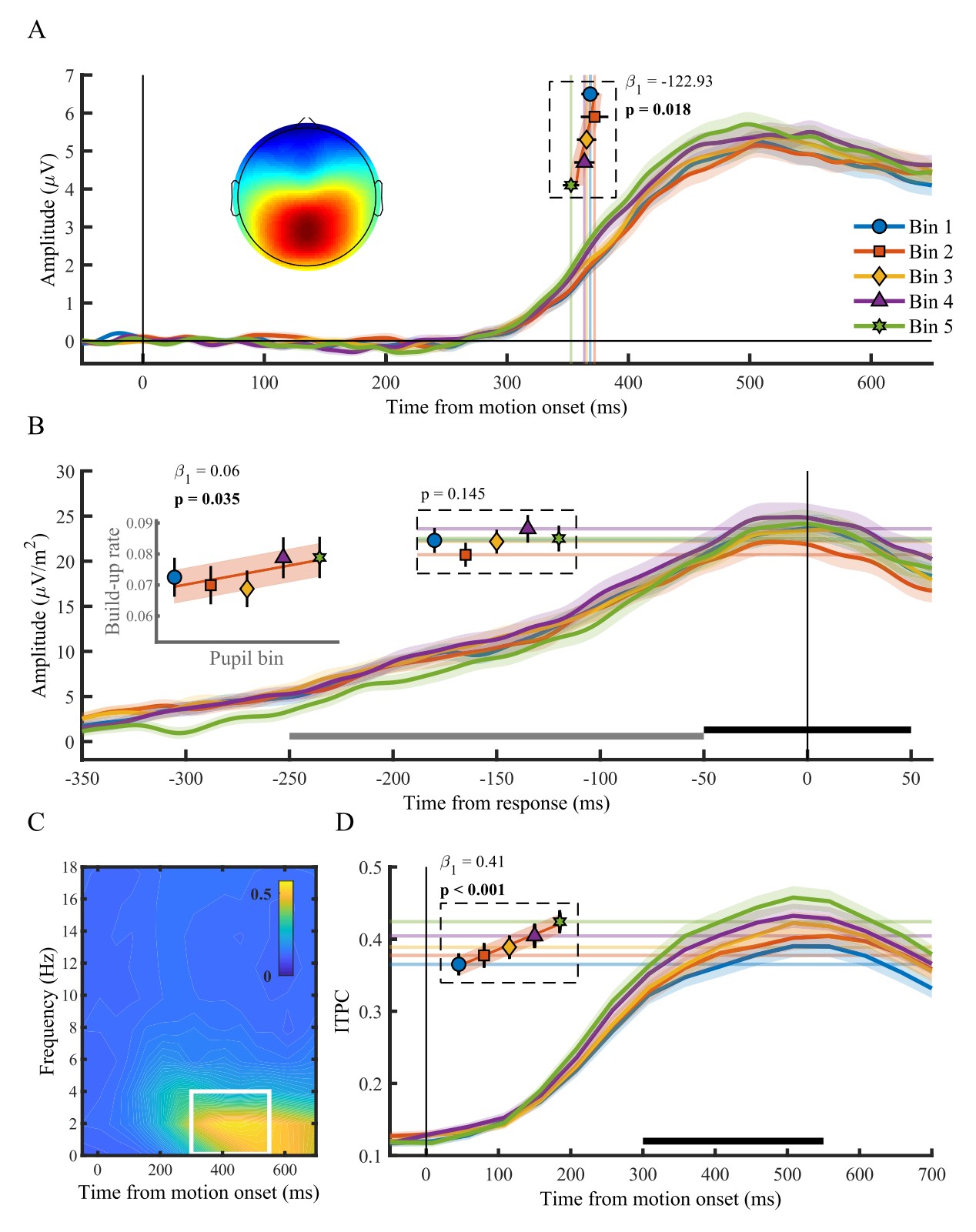

**Figure 2.** The centro-parietal positivity (CPP) in relation to phasic arousal. (**A**) The stimulus-locked CPP time-course shows faster onset times for larger pupil response bins. The inset shows the scalp topography of the CPP. Vertical lines and markers indicate the onset latencies per bin. (**B**) The response-locked CSD-transformed CPP time-course. Horizontal lines and markers indicate the CPP amplitude, and the inset displays the build-up rate of the CPP across pupil response bins. The black bar represents the time window used for the calculation of the CPP amplitude and the grey bar the time window

*Figure 2 continued on next page*

*Figure 2 continued*

used for the calculation of the build-up rate. (C) Grand average inter-trial phase coherence (ITPC) per time-frequency point for the CPP. The white box represents the time-frequency window selected for statistical analyses. (D) ITPC per pupil bin over time for frequencies below 4 Hz. The black bar indicates the time window used for further analysis. Horizontal lines and markers indicate the averaged ITPC in the time-frequency window indicated by the white box in panel C. Error bars and shaded regions denote ±1 SEM. Stats, linear mixed effects model analyses (Statistical analyses). Lines and shading indicate significant fits to the data.

DOI: https://doi.org/10.7554/eLife.42541.012

The following source data and figure supplements are available for figure 2:

**Source data 1.** Csv table containing data for *Figure 2*.

DOI: https://doi.org/10.7554/eLife.42541.015

**Figure supplement 1.** N2c amplitude is reduced when aligned to the response.

DOI: https://doi.org/10.7554/eLife.42541.013

**Figure supplement 2.** CPP ITPC aligned to response.

DOI: https://doi.org/10.7554/eLife.42541.014

(p=0.20) varied with phasic arousal, suggesting that phasic arousal does not influence the build-up rate of choice-related activity over motor cortex.

## Target selection signals do not correlate with the phasic pupil response

Next we investigated the N2 (*Figure 3D–F*), a stimulus-locked early target selection signal that has been shown to predict behavioural performance and modulate the onset and build-up rate of the CPP (*Loughnane et al., 2016*). Because of the spatial nature of the task, we analysed the negative deflection over both the contra- (N2c) and ipsi-lateral (N2i) hemisphere, relative to the target location. The pupil response was not predictive of any aspect of the N2. Specifically, phasic arousal was not predictive of N2c latency (p=0.82) or amplitude (p=0.64), nor did we find any relationship between the pupil response and the N2c ITPC (p=0.14). Likewise, the pupil response was not predictive of N2i latency (p=0.64), amplitude (p=0.11) or ITPC (p=0.87).

## The impact of phasic arousal on task performance is mainly mediated by the consistency in decision processing

We found that pupil-linked phasic arousal was predictive of specific neural signals at multiple information processing stages of perceptual decision making. To test which of these signals explained unique variability in behavioural performance across the five pupil response bins and subjects, the neural signals were added to a linear mixed effects model predicting either RT or RTcv with their order of entry determined hierarchically by their temporal order in the decision-making process. This allowed us to test whether each successive stage of neural processing would improve the fit of the model to the behavioural data, over and above the fit of the previous stage. Compared to the baseline model predicting RT with pupil bin, the addition of pre-target $\alpha$-power significantly improved the model fit ($\chi^2_{(1)}$=10.30, p<0.001). None of the measures of early target selection improved the fit of the model; neither N2c latency ($\chi^2_{(1)}$=0.14, p=0.70) or amplitude ($\chi^2_{(1)}$=0.94, p=0.33), nor N2i latency ($\chi^2_{(1)}$=2.39, p=0.12) or amplitude ($\chi^2_{(1)}$=2.39, p=0.12). We found that both the addition of CPP onset ($\chi^2_{(1)}$=8.24, p=0.004) as well as the build-up rate ($\chi^2_{(1)}$=4.90, p=0.027) significantly improved the model fit. Whereas the addition of CPP amplitude did not ($\chi^2_{(1)}$=1.43, p=0.23), the addition of CPP ITPC substantially improved the fit of the model ($\chi^2_{(1)}$=19.25, p<0.001). Neither the LHB build-up rate nor amplitude improved the fit of the model (LHB build-up rate $\chi^2_{(1)}$=0.02, p=0.88; amplitude $\chi^2_{(1)}$=0.64, p=0.42). Overall, this model suggested that pre-target $\alpha$-power, CPP onset, build-up rate and ITPC exert partially independent influences on RT. Because some variables were highly correlated (e.g. CPP onset and ITPC) we used an algorithm for forward/backward stepwise model selection (*Venables and Ripley, 2002*) to test whether each neural signal indeed explained independent variability that is not explained by any of the other signals. This procedure eliminated CPP onset ($F_{(1)}$ = 0.06, p=0.80) and build-up rate ($F_{(1)}$ = 1.86, p=0.17) from the final model. Thus, only pre-target $\alpha$-power and CPP ITPC significantly improved the model fit for predicting RT. These two variables were forced into one linear mixed effects model predicting RT (Statistical analyses), and comparison to a baseline model revealed a good fit ($\chi^2_{(2)}$=38.61, p<0.001). The fixed effects of the model (the neural signals) explained 15.8% of the variability in RT (marginal $r^2$) across

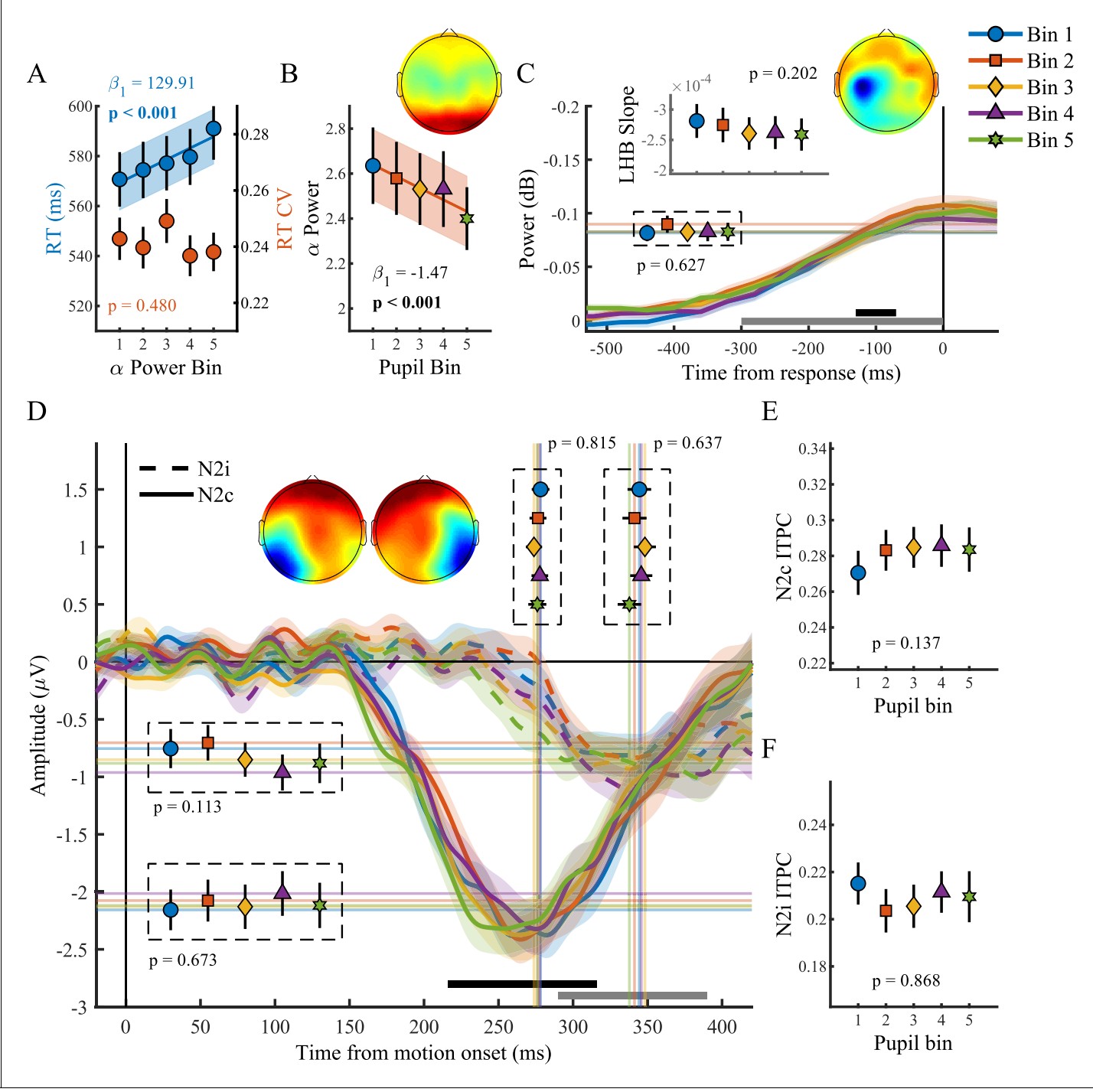

**Figure 3.** The phasic pupil response in relation to EEG signatures of attentional engagement, motor response and early target selection. (A) RT and RTcv in relation to pre-target α power. (B) Pre-target α power in relation to the pupil response. (C) Response-related left hemispheric β power (LHB) per pupil bin. Horizontal lines and markers indicate the average LHB in the time window indicated by the black bar. Inset shows LHB build-up rate, as determined by fitting a straight line through the LHB in the time window indicated by the grey bar. Note the reverse y-axis direction. (D) The stimulus-locked N2c (solid lines) and N2i (dashed lines) time-course binned by the pupil response. Vertical lines and markers show the peak latencies. Horizontal lines and markers show the average N2 amplitude during the time period indicated by the black (N2c) and grey (N2i) bars. (E–F) N2c (E) and N2i (F) ITPC per pupil bin over the time and frequency window determined based on the grand average ITPC (*Figure 3—figure supplement 1*). Insets show the scalp topography of each neural signal. Error bars and shaded regions denote ±1 SEM. Stats, linear mixed effects model analyses (Statistical analyses). Lines and shading indicate significant fits to the data.

DOI: https://doi.org/10.7554/eLife.42541.016

*Figure 3 continued on next page*

*Figure 3 continued*

The following source data and figure supplement are available for figure 3:

**Source data 1.** Csv table containing data for *Figure 3* panel A.
DOI: https://doi.org/10.7554/eLife.42541.017
**Source data 2.** Csv table containing data for *Figure 3* panel B-F.
DOI: https://doi.org/10.7554/eLife.42541.018
**Figure supplement 1.** N2 ITPC.
DOI: https://doi.org/10.7554/eLife.42541.019

the five pupil response bins, and together with the random effects (across subject variability) it explained 92.6% of the variability (conditional $r^2$).

We performed the same hierarchical regression analysis to see which neural signals explained variability in RTcv. We summarised the results of this analysis in *Supplementary file 2*, and report the most important results here. The hierarchical regression analysis revealed that both CPP onset and CPP ITPC improved the model fit, but eliminated CPP onset after the forward/backward model selection. Consequently, CPP ITPC was the only variable that exerted independent influence on RTcv. Comparison against a baseline model revealed a significant fit ($\chi^2_{(1)}$=15.36, p<0.001) that had a marginal $r^2$ of 16.0% and a conditional $r^2$ of 45.9%.

To test whether our assumptions about the temporal order of the neural signals influenced these results, we fitted a model in which all EEG signatures were added at the same time and investigated their coefficients. This analysis did not identify any additional neural components to those that were found using the hierarchical regression analysis (*Supplementary file 3*).

*Table 1* shows the final parameter estimates for the neural signals that significantly predicted variability in RT or RTcv that is due to variability in phasic arousal. From this analysis we can conclude that CPP ITPC was the strongest predictor for RT and the only predictor for RTcv. These results provide novel insight into the mechanism by which the neuromodulators that control arousal can influence behaviour. The impact of these modulators on decision-making is thus mainly mediated by their effects on the consistency in decision formation. Next, we turn to tonic arousal and its relationship to these same EEG components of perceptual decision-making.

## Baseline pupil diameter is inversely related to the consistency of decision processing

*Figure 4* illustrates the relationship between baseline pupil diameter and the CPP. Unlike the pupil response, baseline pupil diameter was not predictive of the onset (p=0.20) or build-up rate (p=0.12), but it displayed an inverse relationship with both the amplitude ($\chi^2_{(1)}$=7.09, p=0.01) and the consistency of the CPP ($\chi^2_{(1)}$=9.34, p=0.002). In line with previous research that revealed increased variability in the rate of evidence accumulation during periods with larger baseline pupil diameter (*Murphy et al., 2014b*), we found an inverse, approximately linear, relationship in which higher baseline pupil diameter displayed lower EEG signal consistency (*Figure 4D*). Thus, states of higher

**Table 1.** Parameter estimates for the final linear mixed effect model of RT and RTcv binned by the pupil diameter response or baseline.
The only parameters included are the neural signals that significantly improved the model fit.

| | RT | | | | RTcv | | | |
|---|---|---|---|---|---|---|---|---|
| | β | β SE | T | P | β | β SE | T | P |
| **Pupil response** | | | | | | | | |
| pre-target α-power | 0.20 | 0.065 | 3.07 | 0.002 | | | | |
| CPP ITPC | −0.19 | 0.034 | −5.51 | <0.001 | −0.21 | 0.049 | −4.22 | <0.001 |
| **Baseline Pupil diameter** | | | | | | | | |
| N2c amplitude | 0.06 | 0.027 | 2.33 | 0.021 | | | | |
| CPP ITPC | −0.17 | 0.033 | −5.21 | <0.001 | −0.31 | 0.056 | −5.48 | <0.001 |

DOI: https://doi.org/10.7554/eLife.42541.020

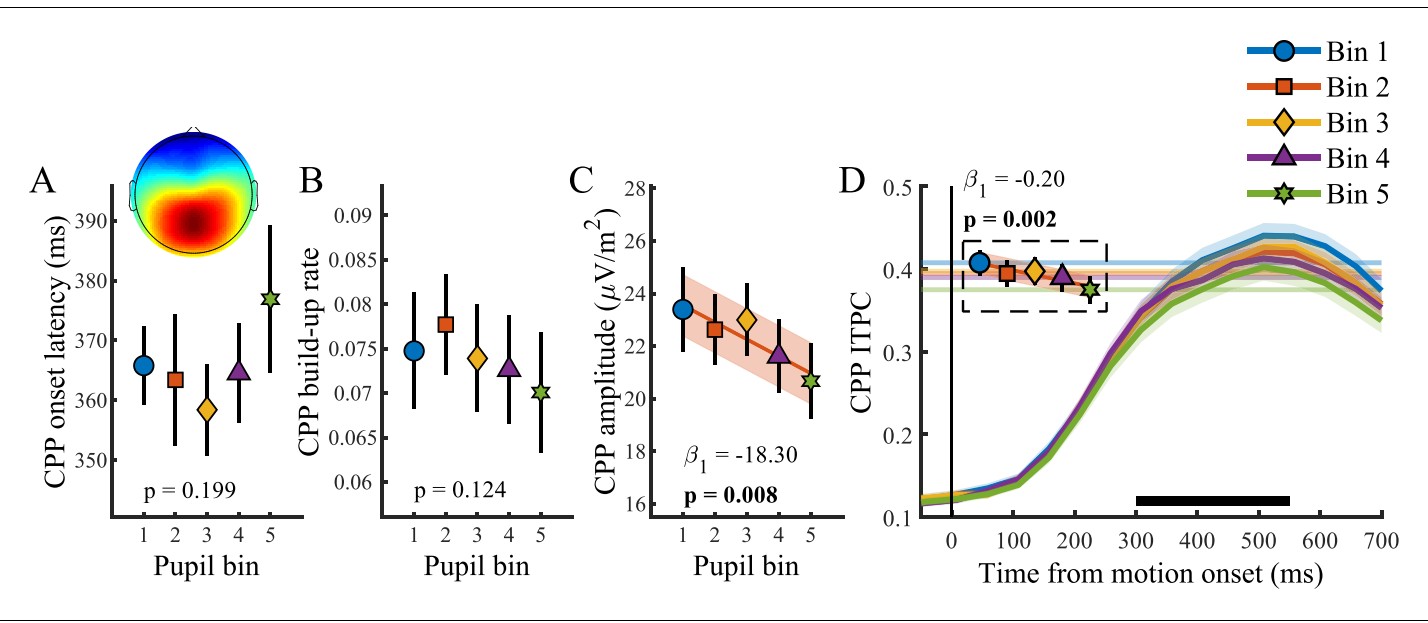

**Figure 4.** Relationship between baseline pupil diameter and the CPP. (**A**) CPP onset latency, (**B**) build-up rate, (**C**) amplitude and (**D**) ITPC per pupil bin over time for frequencies below 4 Hz. The black bar indicates the time window used for further analysis. Horizontal lines and markers indicate the averaged ITPC in the time-frequency window indicated by the white box in *Figure 2C*. Error bars and shaded regions denote ±1 SEM. Stats, linear mixed effects model analyses (Statistical analyses). Lines and shading indicate significant fits to the data.

DOI: https://doi.org/10.7554/eLife.42541.021

The following source data is available for figure 4:

**Source data 1.** Csv table containing data for *Figure 4*.

DOI: https://doi.org/10.7554/eLife.42541.022

arousal are characterized by less consistency, that is more variability, in decision processing. Additionally, states of higher tonic arousal also display lower task performance (*Figure 1C*), indicating that higher variability in decision processing (due to higher tonic arousal) can affect behavioural performance.

## Baseline pupil diameter relates to spectral measures of baseline attentional engagement and motor output as well as early target selection signals

We found a relationship between baseline pupil diameter and specific characteristics of multiple neural processing stages of perceptual decision-making. Specifically, as observed before (*Hong et al., 2014*), pre-target alpha power (*Figure 5A*) varied with baseline pupil diameter in a non-monotonic, but not inverted-U shaped, manner ($\chi^2_{(1)}$=4.49, p=0.034). This suggests that with higher tonic arousal, alpha activity is higher (or less desynchronised). Next, we tested whether baseline pupil diameter was predictive of EEG characteristics representing motor output (*Figure 5B*). We found an inverse relationship with LHB build-up rate ($\chi^2_{(1)}$=10.99, p<0.001), decreasing with larger baseline pupil diameter, but we did not find a relationship with LHB amplitude (p=0.34).

Lastly, we investigated whether baseline pupil diameter affected early target selection signals, the N2 (*Figure 5C–D*). Previous studies have revealed that baseline pupil diameter affected the size and variability of neural responses to visual and auditory stimuli (*Reimer et al., 2014*; *McGinley et al., 2015a*). Here we found that baseline pupil diameter was not predictive of the peak latency of the N2c (p=0.75), but that it did display a monotonic relationship with the N2c amplitude ($\chi^2_{(1)}$=13.72, p<0.001). Trials with larger baseline pupil diameter displayed smaller N2c amplitudes, suggesting that higher arousal has a negative impact on sensory encoding. N2c ITPC did not vary with baseline pupil diameter (p=0.25), and nor did N2i ITPC (p=0.33), N2i latency (p=0.78) or amplitude (p=0.06). We thus found that, similar to the phasic pupil diameter response, baseline pupil diameter is predictive of specific characteristics of each of the processing stages of perceptual

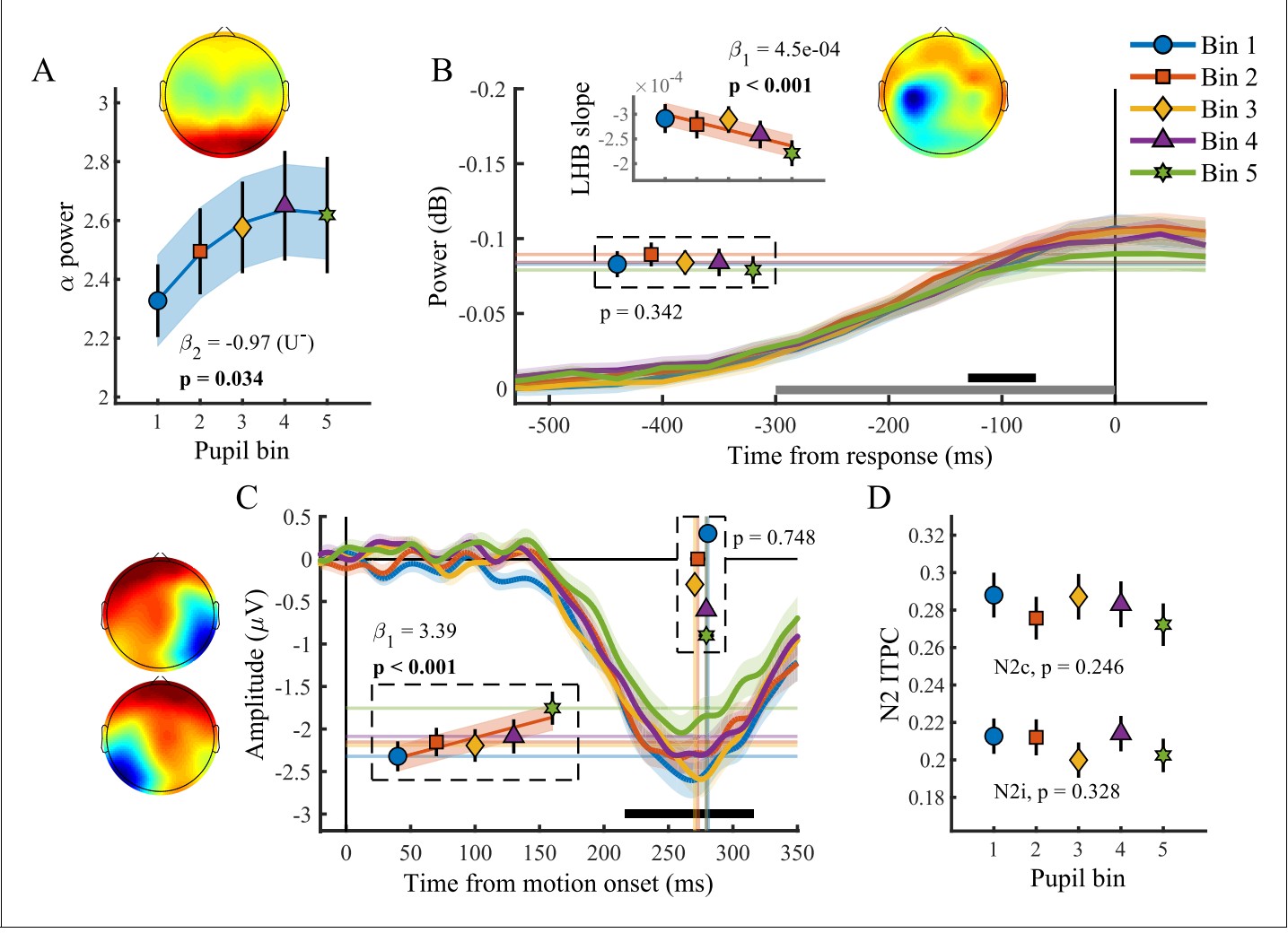

**Figure 5.** Baseline pupil diameter in relation to EEG signatures of attentional engagement, motor response and early target selection. (A) Pre-target α power by baseline pupil diameter. (B) Response-related left hemispheric β power (LHB) per pupil bin. Horizontal lines and markers indicate the average LHB in the time window indicated by the black bar. Inset shows the LHB build-up rate, as determined by fitting a straight line through the LHB in the time window indicated by the grey bar. Note the reverse y-axis direction. (C) The stimulus-locked N2c time-course binned by baseline pupil diameter. Vertical lines and markers show the peak latencies. Horizontal lines and markers show the average N2c amplitude during the time period indicated by the black bar. (D) N2c and N2i ITPC per pupil bin averaged in a time-frequency window based on the grand average (*Figure 3—figure supplement 1*). Insets show the scalp topography of each neural signal. Error bars and shaded regions denote ±1 SEM. Stats, linear mixed effects model analyses (Statistical analyses). Lines and shading indicate significant fits to the data.

DOI: https://doi.org/10.7554/eLife.42541.023

The following source data is available for figure 5:

**Source data 1.** Csv table containing data for *Figure 5*.
DOI: https://doi.org/10.7554/eLife.42541.024

decision-making. Next, we investigated which of these components explained unique variance in task performance across pupil size bins.

## N2c amplitude and CPP ITPC are predictive of variability in task performance due to tonic arousal

We again performed the same hierarchical regression analysis as described above, to see which of the neural signals explained unique variability in task performance associated with tonic arousal. The full results of this analysis are summarised in *Supplementary file 4*. Here we discuss the main findings. After the application of a forward/backward model selection algorithm (*Venables and Ripley,*

*2002*), N2c amplitude and CPP ITPC were the only parameters that were predictive of RT (*Table 1*). These variables were forced into one regression model predicting RT, and comparison against a baseline model with baseline pupil diameter as a factor revealed a significant fit ($\chi^2_{(2)}$=31.6, p<0.001) with a marginal (conditional) $r^2$ of 4.1% (94.4%). This same hierarchical regression procedure revealed that CPP ITPC was the only EEG component that explained unique variability in RTcv (*Table 1*). Comparison against a baseline model also led to a significant fit ($\chi^2_{(1)}$=26.83, p<0.001), with a marginal (conditional) $r^2$ of 11.9% (44.5%). None of the other EEG parameters that were excluded from the final model due to potential false assumptions about their temporal order revealed significant coefficients in a multilevel model analysis in which all components were added simultaneously (*Supplementary file 5*).

Thus, additional to an effect of N2c amplitude on RT, the consistency of the CPP was the only stage of information processing that explained unique within and across-subject variability in task performance associated with changes in baseline pupil diameter.

## Discussion

Here we investigated whether behavioural and neural correlates of decision-making varied as a function of baseline or task-evoked pupil diameter, indexing tonic and phasic arousal, respectively. The perceptual decision-making paradigm employed (*Figure 1A*) allowed us to monitor the relationship between pupil diameter and independent measures of attentional engagement, early target selection, decision formation and motor output. We found that the trial-by-trial variability in both tonic and phasic arousal were predictive of behavioural performance (*Figure 1B & C*). For tonic arousal, this relationship was best described by a non-monotonic polynomial fit with slower RT for higher baseline pupil diameter. Higher phasic arousal, on the other hand, was predictive of better task performance.

We furthermore established that both tonic and phasic arousal were predictive of a subset of EEG signatures, together reflecting discrete aspects of information processing underpinning perceptual decision-making. A hierarchical regression analysis allowed us to determine which of these processing stages exerted an independent influence on behavioural performance associated with central arousal. We found that pre-target α power, indexing baseline attentional engagement, and the consistency of the CPP, reflecting the decision formation, each explained unique variability in task performance that was due to variability in phasic arousal. Variability in task performance due to fluctuations in tonic arousal was explained by the amplitude of the target selection signal N2c and the consistency of the CPP. We thus revealed a direct relationship between both tonic and phasic measures of arousal, and a distinct but overlapping set of EEG signatures of perceptual decision-making, and in particular the CPP.

### The functional significance of the CPP during perceptual decision-making

Although the CPP has previously been found to reflect the accumulation of evidence (*O'Connell et al., 2012*; *Kelly and O'Connell, 2013*; *Loughnane et al., 2016*; *Loughnane et al., 2018*; *Newman et al., 2017*), as discussed in the results section, our task design does not allow us to unequivocally relate the CPP to a specific characteristic of the decision making process such as evidence accumulation. Because of the temporal and spatial uncertainty of the target stimulus, rather than accumulating evidence over an extended period of time on trials with slower RT, target onset transients could be delayed or subjects could be employing different strategies for motion detection on different trials (e.g. verifying the presence of coherent motion in one stimulus versus the other stimulus). Decision signals such as the CPP or LHB could on single trials behave analogously to a step-like signal that across trials seems to be accumulating to a threshold (*Latimer et al., 2015*; but see *Shadlen et al., 2016*), potentially supported by neural mechanisms in V4 that increase their activity transiently in response to changes in motion coherence (*Costagli et al., 2014*). Although we cannot discount that subjects use different strategies on different trials, previous studies in which subjects were required to monitor either one or multiple dot kinematograms revealed no differences in either RT or hit rate (*Loughnane et al., 2016*) and both the early target selection signals and the CPP scaled with the percentage of coherently moving dots (*Loughnane et al., 2016*). We additionally showed here that there is no fixed delay between target selection and

response (*Figure 2—figure supplement 1*) and that there is thus variability in the duration of the sustained period of the task. Any relationship between arousal and the CPP is therefore not solely the result of fluctuations in the latency of the target onset transient.

## Large phasic pupil responses are predictive of better task performance

We estimated the variability in phasic arousal using the amplitude of the task-evoked pupil diameter. Because of the sluggish nature of the pupil diameter response, pupil dilation after target onset likely reflects a combination of specific aspects of phasic arousal such as a response to target onset, decision formation as well as a motor response. Here we aimed to disentangle these different components by applying a general linear model on a single trial basis. First we determined the fit for various models for each subject across trials (*Figure 1—figure supplement 1*), after which we applied the best (across subject) fitting model to each individual trial. We addressed the reliability of the estimation of each of the temporal components by comparing their relationship to behavioural performance to those of other measures of the amplitude of the pupil diameter response (*Figure 1—figure supplement 2*), excluding trials with high VIF values, and orthogonalizing the predictors (*Figure 1—figure supplement 3*), and comparing the results from the single trial parameter estimation to those of groups of trials binned by RT (*Figure 1—figure supplement 4*). These results revealed that we can reliably estimate the target onset component, but that the estimation of the sustained component might not be as straightforward (*Figure 1—figure supplement 3*). Although the current measure of the pupil response to target onset is unlikely to be completely independent of the estimation of the sustained component, inclusion of this predictor increased the fit of the model and captured variability in the pupil time course likely to reflect the influence of phasic arousal specific to decision formation. This reduced the influence of this sustained part of the arousal response on the estimation of the target onset component (*Figure 1—figure supplement 3*). To the extent that we could reliably estimate the amplitude of the target onset component, we investigated its relationship to the behavioural and neural signatures of perceptual decision-making.

Larger target onset responses, presumably reflecting a phasic response in neuromodulatory brainstem centres, were predictive of faster and less variable RT (*Figure 1C*), faster onset, larger build-up rates and higher consistency of the CPP (*Figure 2*), as well as lower pre-target occipital α-power (*Figure 3*). These results can be interpreted in light of the relationship between pupil dilations and the activity in brain areas such as the LC or BF (*Aston-Jones and Cohen, 2005*; *Gilzenrat et al., 2010*; *Varazzani et al., 2015*; *Joshi et al., 2016*; *Reimer et al., 2016*; *de Gee et al., 2017*). Direct electrophysiological recordings from the LC have revealed a positive correlation between LC phasic activity and behavioural performance on elementary target detection tasks (*Aston-Jones et al., 1994*; *Aston-Jones et al., 1997*; *Rajkowski et al., 1994*; *Rajkowski et al., 2004*). Likewise, cue detection is enhanced on trials with a larger cholinergic response (*Parikh et al., 2007*), and previous studies have found that large pupil responses were predictive of higher behavioural performance (*Beatty, 1982b*; but see *Kristjansson et al., 2009*), and decreased decision bias (*de Gee et al., 2017*). Additionally, poor performance upon pupil constrictions is in line with studies showing that sensory target detection is suboptimal when a transient LC or BF response is absent (*Aston-Jones et al., 1994*; *Rajkowski et al., 1994*; *Parikh et al., 2007*; *Gritton et al., 2016*). Moreover, naturally occurring pupillary constrictions are preceded by transient activity decreases in the LC (*Joshi et al., 2016*), and are associated with increased synchronization of cortical activity, a signature of cortical down states, as well as suboptimal processing of visual stimuli (*Reimer et al., 2014*). Our results suggest that event-related pupillary constrictions could be associated with similar neural mechanisms.

Trials with large pupil responses, and better task performance, were preceded by lower pre-target occipital α-power, that is more α desynchronization (*Figure 3*). In line with these results and previous studies (*Kelly and O'Connell, 2013*), lower pre-target α-power itself was predictive of higher task performance. Fluctuations in α synchronization have previously been related to variation in both arousal and attentional deployment (*Ergenoglu et al., 2004*; *van Dijk et al., 2008*; *O'Connell et al., 2009*; *Kelly and O'Connell, 2013*; *Newman et al., 2016*), often interpreted as a neurophysiological correlate of cortical excitability. Here, on trials with both higher phasic arousal and more α desynchronization, behavioural performance was better. This could indicate that fluctuations in phasic arousal and attentional engagement rely on similar neuromodulatory mechanisms.

We additionally found that larger pupil responses were predictive of earlier onset latencies, faster build-up and higher consistency of the CPP signal (*Figure 2*). Thus the effects of the fluctuations in phasic arousal and attentional deployment on task performance are likely mediated by their effect on decision signals, and insofar as the CPP represents evidence accumulation (see above), these fluctuations could influence the build-to-threshold dynamics during perceptual decision-making.

## Large baseline pupil diameter is predictive of relatively poorer task performance

We found a non-monotonic relationship between baseline pupil diameter and task performance (*Figure 1B*). This relationship was, however, not significantly U-shaped, but rather we found slower RT with higher baseline pupil diameter. This effect was moreover only observed when the pupil diameter data was not high-pass filtered (*Figure 1—figure supplement 6*), indicating that slow changes (<0.01 Hz) in pupil diameter are driving the effects on task performance.

In line with previous research (*Hong et al., 2014*), out of all the investigated EEG components, only pre-target α power displayed a small non-monotonic relationship with baseline pupil diameter. Approximately linear relationships were found with N2c amplitude, LHB build-up rate, as well as an inverse relationship with CPP amplitude and ITPC. Of these, only N2c amplitude and CPP ITPC explained within and across subject variability in task performance (*Table 1*). It thus seems that the effects of tonic arousal on task performance are mainly driven by an approximately linear relationship with target selection and consistency of decision formation.

These results appear at odds with a U-shaped relationship as predicted by the adaptive gain theory (*Aston-Jones and Cohen, 2005*), and found during auditory target detection tasks (*Murphy et al., 2011*; *McGinley et al., 2015a*). One potential reason that we did not find a U-shaped relationship with task performance, is that we might not have observed the full range of possible baseline pupil diameter values, and thus not the full range of possible tonic arousal levels. Trials were presented in blocks of 18, after which subjects were allowed to take a short break, preventing them from becoming overly drowsy or too distracted. However, depending on the behavioural paradigm and task demands, the relationship between central arousal, performance and neural activity may take different forms (*McGinley et al., 2015b*). Membrane potential recordings from sensory and association areas, as well as direct electrophysiological recordings from neuromodulatory brainstem centres during decision-making tasks, are needed to gain further insight in the exact mechanisms that drive the relationship between cortical state, sensory encoding, decision formation and task performance.

## Variability in task performance due to pupil-linked arousal is best predicted by the consistency in decision formation

During epochs of quiet wakefulness, membrane potential fluctuations of neurons in visual, somatosensory and auditory cortex are closely tracked by baseline pupil diameter (*Reimer et al., 2014*; *McGinley et al., 2015a*). These fluctuations in subthreshold membrane potential are characteristic of changing cortical state. Small pupil diameter is characterized by prominent low-frequency (2–10 Hz) and nearly absent high-frequency oscillations (30–80 Hz), whereas larger pupil diameter is characterized by reduced low-frequency, but increased high-frequency oscillations (*McGinley et al., 2015a*; *McGinley et al., 2015b*). Thus, the average subthreshold membrane potential is most stable during intermediate pupil diameter, when neither low nor high-frequency components predominate. States of lower variability are furthermore characterized by more reliable sensory responses, higher spike rates, increased neural gain and better behavioural performance (*Reimer et al., 2014*; *McGinley et al., 2015a*; *McGinley et al., 2015b*). In addition to activity in early sensory areas, there is some evidence that activity in higher-order association areas is also more reliable with intermediate arousal. During auditory target detection, human subjects displayed the least variable RT at intermediate baseline pupil diameter, as well as the highest amplitudes of the P3 component elicited by task-relevant stimuli (*Murphy et al., 2011*).

Here we found that the consistency of the CPP was the main EEG predictor of variability in task performance associated with both tonic and phasic arousal. For tonic arousal, our findings are largely in line with modelling studies which suggested that higher arousal is specifically predictive of more variability in evidence accumulation (*Murphy et al., 2014b*). For phasic arousal, higher consistency,

and thus less variability, was found for larger pupil bins, which also displayed the best behavioural performance. These results suggest that similar neural mechanisms of cortical state described for sensory cortex (*Reimer et al., 2014*; *McGinley et al., 2015b*; *McGinley et al., 2015a*; *Vinck et al., 2015*) might also affect neurons in higher-order association areas (e.g. parietal cortex) and thereby influence evidence accumulation and task performance. Simultaneous pupil diameter and membrane potential recordings in parietal cortex during decision-making are needed to confirm this hypothesis.

## Target selection signal amplitude is modulated by pupil-linked arousal

In the present study, we used a paradigm in which two stimuli were continuously presented and target occurrence was both spatially and temporally unpredictable. Successful target detection thus relied on locating and selecting sensory evidence from multiple sources of information. *Loughnane et al. (2016)* have shown that early target selection signals, which occur contralateral to the target stimulus (N2c), modulate sensory evidence accumulation and behavioural performance. Although previous studies have characterised the dependence of the quality of sensory responses on fluctuations in cortical state, as measured by baseline pupil diameter (*Reimer et al., 2014*; *McGinley et al., 2015a*; *Vinck et al., 2015*), to the best of our knowledge, the influence of pupil-linked arousal on target selection signals has not been described before. Here, we showed that early target selection signals are modulated by tonic arousal such that larger baseline pupil diameter was predictive of smaller N2c amplitudes (*Figure 5C*). Moreover, the amplitude of the N2c also explained unique variability in task performance across pupil bins and subjects (*Table 1*).

At first glance it seems counterintuitive that target selection signal amplitudes are decreased, whereas visual encoding in early visual cortex is enhanced on trials with larger baseline pupil diameter (*Vinck et al., 2015*), or during pupil dilation (*Reimer et al., 2014*). These differences could be due to differences in the nature of the recordings, as these previous studies used invasive electrophysiology and calcium imaging whereas we used scalp EEG, limiting especially the spatial resolution of our analyses that might be necessary to elucidate these effects (e.g. single neuron orientation tuning). Alternatively, they could constitute differential effects of arousal on visual encoding and target selection. More likely, however, they are due to specific task demands, in particular our use of multiple simultaneously presented competing stimuli. Indeed, there is some evidence that an increase in arousal, as measured by pupil diameter, can increase the ability of a distractor to disrupt performance on a Go/No-Go task in non-human primates (*Ebitz et al., 2014*). At high arousal levels, performance might thus be negatively affected when the task requires the successful suppression of distracting information, that is with higher arousal it is more difficult to focus on the task at hand (*Aston-Jones and Cohen, 2005*; *McGinley et al., 2015b*). On the current task, it might thus be more difficult to select and process information from one of the two competing stimuli during states of high arousal, leading to reduced N2c amplitude as well as reduced performance.

## The overlap and dissociation between baseline pupil diameter and the pupil response

As in previous studies (*Gilzenrat et al., 2010*; *Murphy et al., 2011*; *de Gee et al., 2014*), we found a negative correlation between baseline pupil diameter and the size of the pupillary response. Both measures were predictive of task performance as well as a unique, but overlapping, set of EEG signatures of perceptual decision-making. Because of the overlap in their effects on these EEG markers, in particular pre-target α power and CPP ITPC, it is possible that both (in part) reflect the same component of central arousal state. Although we removed (via linear regression) the variance in the pupil response that is due to fluctuations in the amplitude and the phase of the baseline pupil diameter, some variability in the baseline pupil diameter might not be fully dissociable from the pupil response, and both might thus reflect a noisy measure of tonic arousal. This interpretation is further supported by the finding that the relationship between the pupil response and task performance did not substantially change regardless of whether variability in the pupil response due to fluctuations in baseline amplitude and/or phase was removed or not (*Figure 1—figure supplement 5*). Importantly, however, the dissociation in the effect of baseline pupil diameter and the pupil response on these EEG markers, such as the effect on N2c amplitude, indicates that these measures also capture

independent variability in central arousal (tonic and phasic) predictive of distinct information processing stages of decision-making.

## Concluding remarks

In this study we investigated the relationship between measures of tonic and phasic pupil-linked arousal and behavioural and EEG measures of perceptual decision-making. We found that trial-to-trial variability in both tonic and phasic arousal accounted for variability in task performance and were predictive of a unique, but overlapping, set of neural metrics of perceptual decision-making. Specifically, tonic arousal exerted its influence on task performance through its effects on early target selection signals and the consistency of decision formation. Phasic arousal, on the other hand, affected behaviour through its relation with attentional engagement as well as the consistency of decision formation. These results indicate that during decision-making both tonic and phasic activity in the (network of) neuromodulatory centres that control central arousal can affect behaviour during perceptual decision-making. Thus, fluctuations in central arousal, mediated by neuromodulatory brainstem centres, act on multiple timescales to influence task performance through its effects on attentional engagement, sensory processing as well as decision formation.

## Materials and methods

### Task procedures

Subjects (n = 80) and methods are largely overlapping with the details and procedures described elsewhere (*Newman et al., 2017*). Here we summarise details necessary to understand this study, and we also describe procedures that differ from the previous study. Participants were seated in a darkened room, 56 cm from the stimulus display (21 inch CRT monitor, 85 Hz 1024 × 768 resolution), asked to perform a continuous bilateral variant (*O'Connell et al., 2012*; *Kelly and O'Connell, 2013*) of the random dot motion task (*Newsome et al., 1989*; *Britten et al., 1992*). Subjects fixated on a central dot while monitoring two peripheral patches of continuously presented randomly moving dots (*Figure 1A*). At pseudorandom times, an intermittent period of coherent downward motion (50%) occurred in either the left or the right hemifield. Upon detection of coherent motion, participants responded with a speeded right-handed button press. A total of 288 trials were presented over 16 blocks (18 trials per block). Data were collected under identical experimental procedures at either Monash University, Australia, or Trinity College Dublin, Ireland. The experimental protocol was approved by each University's human research ethics committee before testing (Project number Monash University: 3658, Trinity College: SPREC012014-1), and carried out in accordance with approved guidelines. Informed consent was obtained from all participants before testing.

### Data acquisition and preprocessing

Electroencephalogram (EEG) was recorded from 64 electrodes using an ActiveTwo (Biosemi, 512 Hz) system at Trinity College Dublin, Ireland or a BrainAmp DC (Brainproducts, 500 Hz) at Monash University, Australia. Data were processed using both custom written scripts and EEGLAB functions (*Delorme and Makeig, 2004*) in Matlab (MathWorks). Noisy channels were interpolated after which the data were notch filtered between 49–51 Hz, band-pass filtered (0.1–35 Hz), and rereferenced to the average reference. Data recorded using the Biosemi system were resampled to 500 Hz and combined with the data recorded with the Brainproducts system. Epochs were extracted from −800 to 2800 ms around target onset and baselined with respect to −100 to 0 ms before target onset. To minimize volume conduction and increase spatial specificity, for specific analyses the data were converted to current source density (*Kayser and Tenke, 2006*). We rejected trials from analyses if the reaction times were <150 or>1700 ms after coherent motion onset, or if either the EEG on any channel exceeded 100 mV, or if the subject broke fixation or blinked (Pupillometry) during the analysis period of the trial, the 500 ms preceding target onset (26.59 ± 2.94) for pre-target α power activity or the interval of 100 ms before target onset to 200 ms after the response (33.66 ± 3.95).

Pre-target α-band power (8–13 Hz), N2 amplitude and latency, CPP onset and build-up rate and response related β-power amplitude and build-up rate were computed largely in the same way as in *Newman et al., 2017*. Briefly, α-band power was computed over the 500 ms preceding target onset using temporal spectral evolution (TSE) methods (*Thut et al., 2006*), and pooled over two

symmetrical parietal regions of interest, using channels O1, O2, PO3, PO4, PO7 and PO8. The N2 components were measured at electrodes P7 and P8, ipsi- and contralateral to the target location (*Loughnane et al., 2016*; *Newman et al., 2017*), and the CPP was measured at central electrode Pz. These signals were aggregated to an average waveform for each pupil bin and each participant. We determined the latency of the N2c/N2i as the time point with the most negative amplitude value in the stimulus-locked waveform between 150-400/200-450 ms, while N2c/N2i amplitude was measured as the mean amplitude inside a 100 ms window centered on the stimulus-locked grand average peak (266/340 ms) (*Loughnane et al., 2016*; *Newman et al., 2017*).

Onset latency of the CPP was measured by performing running sample point by sample point t-tests against zero across each participant's stimulus-locked CPP waveforms. CPP onset was defined as the first point at which the amplitude reached significance at the 0.05 level for $\geq$ 15 consecutive points. Because we decreased our statistical power by binning the trials into five bins (see pupillometry), we did not find an onset for every bin for a subset of subjects (baseline pupil diameter: 13 bins over 11 subjects, pupil response: 16 bins over 12 subjects). Because of our use of linear mixed effect analyses, these subjects could still be included in the analysis, with only the missing values being omitted. Both CPP build-up rate and amplitude were computed using the response-locked waveform of the CSD transformed data to minimize influence from negative-going fronto-central scalp potentials (*Kelly and O'Connell, 2013*). Build-up rate was defined as the slope of a straight line fitted to this signal in the window from $-250$ ms to $-50$ ms before response. CPP amplitude was defined as the mean amplitude within the 100 ms around the response.

Response related left hemisphere β-power (LHB, 20–35 Hz) was measured over the left motor cortex at electrode C3 using short-time Fourier transform (STFT) with a 286 ms window size and 20 ms step size (*O'Connell et al., 2012*; *Newman et al., 2017*). We baselined LHB using an across-trial baseline for each subject. LHB amplitude was measured from the response-locked waveform in the window from $-130$ to $-70$ ms preceding the response, whereas the LHB build-up rate was defined as the slope of a straight line fitted to this same waveform in the 300 ms before the response.

Inter-trial phase coherence (ITPC) was estimated using single-taper spectral methods from the Chronux toolbox (*Bokil et al., 2010*) and adapted scripts. We used a 256 sample (512 ms) sliding short time window, with a step size of 25 samples (50 ms). This gave us a half bandwidth (W) of 1.95 Hz: W = (K + 1)/2T, with K being the number of data tapers, K = 1, and T (s) being the length of the time window. Frequencies were estimated from 0.1 to 35 Hz.

## Pupillometry

Eye movements and pupil data were recorded using an SR Research EyeLink eye tracker (Eye-Link version 2.04, SR Research/SMI). Automatically identified blinks and saccades were linearly interpolated from 100 ms before to 100 ms after the event, the interpolated pupil data was then low-pass (<6 Hz) or band-pass (0.01–6 Hz) filtered (second order butterworth). The instantaneous phase of the pupil diameter was calculated by taking the angle of the analytic signal acquired by using the Hilbert transform of the filtered data. Epochs were extracted from $-800$ to 4800 ms around coherent motion onset. Trials in which fixation errors or blinks occurred within the analysis period, from 100 ms before target onset to 200 ms after response, were excluded from analysis. Fixation errors were defined as gaze deviations of more than 3°. The pupil diameter was normalized by dividing by the maximum pupil diameter on any trial in the analysis window from 100 ms before target onset to 200 ms after the response for each subject, and baselined on a single trial basis. We computed the baseline pupil diameter by averaging the pupil diameter in the 100 ms before target onset, and the baseline phase was calculated as the average phase angle in the 100 ms preceding target onset.

We identified the shape of the neural input to the pupil system by applying various general linear models (GLM) to the pupil time course (*Hoeks and Levelt, 1993*; *de Gee et al., 2014*; *Murphy et al., 2016*) with two temporal components corresponding to target and response onset (all models), and a third sustained component (models 2–9) for which the shape varied across eight candidate models tested previously (Figure 5 in *Murphy et al., 2016*). In model 1, only the stimulus and response onset were modelled. The sustained component in the remaining models took the shape of: (model 2) a boxcar component with a constant amplitude throughout the decision interval; (model 3) a linear up-ramp that grew in amplitude with increasing decision time; (model 4) a ramp-to-threshold; (model 5) a linear decay with a starting amplitude that was larger for slower RTs but whose amplitude always terminated at zero; (model 6) a linear decay-to-threshold which began at a

fixed amplitude and terminated at zero; (model 7–9) versions of the boxcar, up-ramp and down-ramp models in which the sustained component was normalized by the number of the samples in that trial's decision interval. We convolved these onset, response and/or sustained temporal components with a pupil impulse response function (IRF):

$$h(t) = t^w e^{-t(w/t_{max})}$$

where $w$ is the width (10.1) and $t_{max}$ is the time-to-peak (930 ms) of the IRF (*Hoeks and Levelt, 1993*; *de Gee et al., 2014*; *Murphy et al., 2016*). Each model was regressed onto the concatenated band-pass filtered pupil diameter time series (from -800 ms before target onset to 2500 ms after the response). Bayes information criterion (BIC) was used to assess model fit:

$$\mathrm{BIC} = n + n\log(2\pi) + \log(SSR/n) + (k+1)\log(n)$$

where $n$ is the number of samples, SSR is the residual sum of squares, and $k$ is the number of free parameters. The goodness of fit between any two models was assessed non-parametrically by applying Wilcoxon signed rank tests to their difference score. We found that the linear up-ramp model (model 3) provided the best fit to the data. *Figure 1—figure supplement 1* illustrates the relative goodness-of-fit of each model, compared to the best fitting linear up-ramp model, as well as the effect size of each of the components of the linear up-ramp model.

To investigate the relationship between pupil-linked arousal and behavioural performance during decision-making, we binned our behavioural and EEG data according to either the baseline pupil diameter or the post target pupil response (see below) into five equally sized bins (mean 49.63 ± SEM 0.81 trials, minimum bin size = 20 trials) (*Figure 1B & C*). The division into five bins allowed us to investigate possible quadratic trends in the data. We used linear regression to remove the trial-by-trial fluctuations in single-trial pupil amplitudes that could be due to inter-trial interval, target side, as well as baseline pupil diameter amplitude or phase, all factors that are known to influence either the post target pupil response and/or behavioural response times (*Kristjansson et al., 2009*; *de Gee et al., 2014*; *Kloosterman et al., 2015*; *Newman et al., 2017*). To partial out the effect of phase, a circular variable, we used the sine and cosine of the phase as orthogonal, linear predictor variables (*Fisher, 1993*). To verify the (absence of) correlation between pupil baseline phase and response before and after the regression, we made use of the circstat toolbox (*Berens, 2009*).

We estimated the task-evoked phasic arousal response according to various single-trial scalar measurements of the amplitude of the pupil response (*Figure 1—figure supplement 2*). The relationship between the average pupil diameter around RT and behavioural performance was best described by a non-monotonic, U-shaped, relationship (*Figure 1—figure supplement 2A*). Because of the temporal low-pass characteristics of the peripheral pupil system (*Hoeks and Levelt, 1993*), trial-to-trial variation in RT can affect the measurement of the size of the pupil response. To remove the trial-to-trial fluctuations in pupil responses due to variations in RT, we removed these components via linear regression (*de Gee et al., 2017*; *Urai et al., 2017*). After the elimination of the contribution of RT to the pupil response, we still observed a U-shaped relationship with behavioural performance (*Figure 1—figure supplement 2B*). This measure of the pupil response, however, likely reflects both the transient response to target onset as well as any activity that occurs thereafter (e.g. during decision formation). Therefore, we aimed to isolate activity specific to the phasic response to target onset. To this end, we computed the mean, slope and linear projection (*de Gee et al., 2014*; *Kloosterman et al., 2015*) over a 400 ms time window around the peak of the derivative of the pupil IRF (636 ms using the canonical IRF). A time-window in which activity occurring after the target onset transient is, presumably, not yet reflected. We found an inverse relationship between each of these measures and behavioural performance (*Figure 1—figure supplement 2C–E*), with better behavioural performance for larger pupil responses. Although these results suggest that measurements of the pupil response in this time-window reflect a different component of the neural input to the pupil system than the measurements of the amplitude around RT (*Figure 1—figure supplement 2A & B*), the use of any specific time window can be interpreted as arbitrary.

To further disentangle the pupil response into separate temporal components, we applied the best-fitting GLM, the linear up-ramp model (*Figure 1—figure supplement 1*), to individual trials by considering each individual trial as a separate condition (*Bach et al., 2018*). Because we reduced the

amount of data used for the regression analysis by applying it to single-trial data, we tested whether this led to collinearity amongst the temporal components by computing the variance inflation factor (VIF). Although large VIF values do not necessarily imply that no conclusions can be drawn from regression analysis (*O'brien, 2007*), as a rule of thumb, VIF values larger than 5 or 10 indicate that predictors are collinear (*Sheather, 2009*; *James et al., 2017*). When applying the GLM across all trials, the average VIF values are within the range of collinearity (*Figure 1—figure supplement 3A–B*). When we apply the same model to single trial data, however, the average VIF values are substantially higher (*Figure 1—figure supplement 3C–D*). It seemed particularly problematic to reliably estimate the sustained and the response component as their VIF scores are larger than 10. The target onset component, on the other hand, has an average VIF score of approximately 5. Single-trial VIF estimates larger than five for target onset ($39.34 \pm 2.84\%$ of trials) were mainly found on trials with short RT (*Figure 1—figure supplement 3E*), revealing that it is difficult to distinguish between these temporal components on short trials. The overall results were, however, not affected by these trials. Repeating the analysis when excluding trials with VIF values larger than five revealed the same relationship pattern between pupil response amplitude and behavioural performance (*Figure 1—figure supplement 3F&G*). Sorting the pupil diameter according to the estimate of the amplitude of the sustained component, revealed that the largest sustained component occurred on trials with a small (or absent) response to target onset (*Figure 1—figure supplement 3H*). Rather than solely reflecting phasic arousal during decision formation, the presence of the sustained component could, for instance, indicate a compensatory mechanism for the absence of an early target onset transient. As the relationship with behavioural performance followed a downward trend when plotted against the target onset component (*Figure 1C*), and an upward trend when plotted against the sustained component (*Figure 1—figure supplement 3I*), together these effects could explain the U-shaped relationship between behavioural performance and the pupil response when measured as the average pupil diameter around RT (*Figure 1—figure supplement 2A–B*).

Although a target-response onset only model was the worst fitting model across trials (*Figure 1—figure supplement 1*), we tested whether a target-response only model could reliably estimate the single-trial target-onset response amplitude. The relationship between this component and behavioural performance (*Figure 1—figure supplement 3K&L*), however, strongly resembled the U-shaped relationship between behaviour and the pupil response amplitude when calculated as the mean amplitude around RT (*Figure 1—figure supplement 2A*), a measure likely to be confounded by both RT and the neural input that occurs after the target onset transient. This supports the notion that the inclusion of a sustained component can make the estimation of the target onset component (amongst others) more accurate, despite the potential collinearity of these predictors. Indeed, the difference in model fit ($R^2$) is significantly larger than 0 for each individual subject (one-sided Wilcoxon signed rank test, data not shown).

*Figure 1—figure supplement 3J* illustrates the average difference in $R^2$ values between single-trial models with and without the sustained component. Lastly, *Figure 1—figure supplement 3M&N* illustrate the actual pupil diameter time course and the single trial fitted pupil diameter, revealing that this model is able to capture considerable variability in the pupil diameter trace.

Next, we applied the same linear up-ramp model to five subsets of trials, binned by RT (average bin size: $50.01 \pm 0.82$ trials). This analysis revealed that the relationship between RT bin and the estimated amplitude of the target component (*Figure 1—figure supplement 4A*) follows a pattern that is highly similar to the relationship between single-trial estimates of the phasic pupil response to target onset and RT (*Figure 1C*), further supporting the notion that the single-trial GLM approach can accurately estimate the target onset transient. We again investigated the VIF values for each of the temporal components of the model applied to the binned data. Although the sustained and response components displayed relatively large values, the target onset component was smaller than 5. Again, large VIF values by themselves are not necessarily cause for concern, if a regression coefficient is statistically significant, even when its VIF value is large, it is significant 'in the face of that collinearity' (*O'brien, 2007*). To further exclude the possibility that large VIF values brought about these results, we repeated this analysis using the data binned according to RT in three or two bins (*Figure 1—figure supplement 4B–C*). These analysis also revealed smaller target onset component coefficients for larger RT, with progressively lower VIF values. Finally, we investigated the relationship between RT and the target onset component after Gram-Schmidt orthogonalization of the predictors (*Figure 1—figure supplement 4D–E*), which eliminated collinearity amongst the temporal

components. After orthogonalization, we again found that the estimate of the β weights of the target onset component was inversely related to RT, both when estimated across bins of trials (*Figure 1—figure supplement 4D*) as well as when estimating this component on a single trial basis (*Figure 1—figure supplement 4E*).

Altogether, these analyses reveal that although the estimation of different temporal components contributing to a single-trial pupil diameter time course has to be done with caution, in the context of the various measures of the phasic pupil response (*Figure 1—figure supplement 2*) and the interpretation of VIF factors (*Figure 1—figure supplement 3* & *Figure 1—figure supplement 4*), it is possible (in this dataset) to extract meaningful estimates of the target onset component.

## Statistical analyses

We used RStudio (RStudio Team (2016). RStudio: Integrated Development for R. RStudio, Inc., Boston, MA URL http://www.rstudio.com) with the package *lme4* (*Bates et al., 2015*) to perform a linear mixed effects analysis of the relationship between baseline pupil diameter or the pupil response and behavioural measures and EEG signatures of detection. As fixed effects, we entered pupil bin (see Pupillometry) into the model. As random effects, we had separate intercepts for subjects, accounting for the repeated measurements within each subject. We sequentially tested the fit of a monotonic relationship (first-order polynomial) against a baseline model (zero-order polynomial), and a non-monotonic (second-order polynomial) against the monotonic fit by means of maximum likelihood ratio tests, using orthogonal polynomial contrast attributes. The behavioural or EEG measure $y$ was modelled as a linear combination of polynomial basis functions of the pupil bins ($X$):

$$y \sim \beta_0 + \beta_1 X + \beta_2 X^2$$

with $\beta$ as the polynomial coefficients. This multilevel approach was preferred over a standard repeated measures analysis of variance (ANOVA), because it allowed us to test for first and second-order polynomial relationships, as well as to account for missing values in the CPP onset estimation. We used a variant of the 'two-lines' approach (*Simonsohn, 2017*), to test for the presence of (inverted) U-shape relationships when a second-order polynomial best fit the data. Using the same multilevel model, we fit two straight lines to the first and last set of two/three bins. For a non-monotonic relationship to be classified as U-shaped, both components needed to have significant coefficients of opposite sign. We iteratively tested the first 3 against the last 2, the first 2 against the last 3 or the first 2 against the last 2 bins (omitting the middle bin), stopping if both criteria were met (p < 0.05, Bonferroni corrected).

To verify that the relationship between pupil diameter and task performance was not dependent on the binning procedure, we ran another regression analysis wherein we predicted single trial RT by sequentially adding the linear and quadratic coefficients for baseline pupil diameter (*BPD*) and pupil response (*PR*):

$$RT \sim \beta_0 + \beta_1 BPD + \beta_2 BPD^2 + \beta_3 PR + \beta_4 PR^2$$

with $\beta$ as the polynomial coefficients. We compared the first model to a random-intercept-only model including subject ID, inter-trial interval, stimulus side, as well as the trial and block number (to control for potential time on task effects), and tested the fit of subsequent models to the previous model fit. This analysis revealed a significant improvement for each step of the sequential analysis, for which the results and parameters estimates are shown in *Supplementary file 1*. These analyses confirm that both the size of the baseline pupil diameter and the pupil response are predictive of task performance on a single trial basis. This relationship moreover follows a non-monotonic, quadratic, function.

After testing the relationship between behavioural and neural signatures of decision-making and pupillometric measures individually, the neural signals were added sequentially into consecutive regression models predicting RT and RTcv. This model had both a random intercept for each subject, allowing for different baseline-levels of behavioural performance, as well as a random slope of pupil bin for each subject, which allowed for across-subject variation in the effect of pupil bin on behavioural performance. The hierarchical entry of the predictors allowed us to model the individual differences in behavioural performance, as a function of the EEG signals representing each temporal stage of neural processing. Starting with preparatory signals (α-power), to early target selection

signals (N2), to evidence accumulation (CPP), to motor preparation (LHB). The hierarchical addition of the predictors informed us whether each of the EEG signals reflecting successive stages of neural processing improved the fit of the model predicting behavioural data. The signals that explained unique variance were then simultaneously forced into a simplified model predicting RT or RTcv, which made it possible to obtain accurate parameter estimates not contaminated by signals that were shown not to improve model fits. Note that only subjects for which we could determine the CPP onset latency for all bins were included in this hierarchical model. For this final model, all behavioural and neural variables were scaled between 0 and 1 across subjects according to the formula: $y_i = (x_i - \min x_i)/(\max x_i - \min x_i)$, where $y_i$ is the scaled variable, $x_i$ is the variable to be scaled. This scaling procedure did not change the relationship of the variable within or across subjects, but scaled all predictor variables to the same range. Again, significance values were obtained by means of maximum likelihood ratio tests.

Data plotted in all figures are the mean and the standard error of the mean (SEM) across subjects. Linear fits are plotted when first-order fits were superior to the zero-order (constant) fit, quadratic fits are plotted when second-order fits were superior to the first-order fit.

## Notes

Raw data (https://figshare.com/s/8d6f461834c47180a444) are open access and available under a Creative Commons Attribution-NonCommercial-ShareAlike 3.0 International Licence. Analysis scripts are freely available on github (*van Kempen, 2019*; copy archived at https://github.com/elifescien-ces-publications/2019_pupil_decisionMaking).

## Acknowledgements

We thank Peter R Murphy for advice on data analysis. JK and AT are supported by research funding from the Henry Wellcome Trust (093104). MAB is supported by fellowship and project grant support from the Australian Research Council (ARC; FT130101488; DP150100986; DP180102066). AT and MAB are supported by research funding from a strategic research partnership between the Newcastle University and Monash University. MAB, ROC and AT are supported by research funding from the Office of Naval Research Global (ONR Global).

## Additional information

### Funding

| Funder | Grant reference number | Author |
| --- | --- | --- |
| Wellcome | 093104 | Jochem van Kempen<br>Alexander Thiele |
| Australian Research Council | FT130101488 | Mark A Bellgrove |
| Office of Naval Research Global | | Alexander Thiele<br>Redmond G O'Connell<br>Mark A Bellgrove |
| Newcastle University, Monash University | | Alexander Thiele<br>Mark A Bellgrove |
| Australian Research Council | DP150100986 | Mark A Bellgrove |
| Australian Research Council | DP180102066 | Mark A Bellgrove |

The funders had no role in study design, data collection and interpretation, or the decision to submit the work for publication.

### Author contributions

Jochem van Kempen, Conceptualization, Formal analysis, Investigation, Visualization, Methodology, Writing—original draft, Writing—review and editing; Gerard M Loughnane, Conceptualization, Data curation, Methodology, Writing—review and editing; Daniel P Newman, Conceptualization, Data curation, Methodology; Simon P Kelly, Supervision, Writing—review and editing; Alexander Thiele,

Supervision, Funding acquisition, Writing—review and editing; Redmond G O'Connell, Conceptualization, Resources, Supervision, Funding acquisition, Writing—review and editing; Mark A Bellgrove, Conceptualization, Resources, Supervision, Funding acquisition, Writing—original draft, Writing—review and editing

**Author ORCIDs**
Jochem van Kempen http://orcid.org/0000-0002-0211-9545
Gerard M Loughnane http://orcid.org/0000-0003-1961-5294
Daniel P Newman http://orcid.org/0000-0001-8240-1876
Simon P Kelly http://orcid.org/0000-0001-9983-3595
Alexander Thiele https://orcid.org/0000-0003-4894-0213
Redmond G O'Connell http://orcid.org/0000-0001-6949-2793
Mark A Bellgrove http://orcid.org/0000-0003-0186-8349

**Ethics**
Human subjects: The experimental protocal was approved by the human research ethics committee from Monash University and Trinity College Dublin, and informed consent was obtained from all participants before testing. Project number Monash University: 3658, Trinity College: SPREC012014-1

**Decision letter and Author response**
Decision letter https://doi.org/10.7554/eLife.42541.032
Author response https://doi.org/10.7554/eLife.42541.033

# Additional files

## Supplementary files

• Supplementary file 1. Parameter estimates for the single-trial mixed effect model analysis predicting RT using linear and polynomial basis functions of baseline pupil diameter (BPD) and the pupil response (PR).
DOI: https://doi.org/10.7554/eLife.42541.025

• Supplementary file 2. Results from model comparisons of the hierarchical regression analysis predicting variability in task performance due to phasic arousal. Boldface font indicates parameters that significantly improved the model fit compared to the addition of the neural signal associated with the previous neural processing stage. Red text indicates the parameters that were excluded from the final model during the forward/backward stepwise regression (main text). Final model fits revealed a marginal (conditional) $r^2$ of 15.8% (92.6%) and 16.0% (45.9%) for RT and RTcv, respectively.
DOI: https://doi.org/10.7554/eLife.42541.026

• Supplementary file 3. Coefficients from the multilevel model analysis in which all EEG components were added simultaneously to predict variability in task performance due to variability in phasic arousal.
DOI: https://doi.org/10.7554/eLife.42541.027

• Supplementary file 4. Results from model comparisons of the hierarchical regression analysis predicting variability in task performance due to tonic arousal. Boldface font indicates parameters that significantly improved the model fit compared to the addition of the neural signal associated with the previous neural processing stage. Red text indicates the parameters that were excluded from the final model during the forward/backward stepwise regression (main text). Final model fits revealed a marginal (conditional) $r^2$ of 4.2% (94.4%) and 11.9% (44.5%) for RT and RTcv, respectively.
DOI: https://doi.org/10.7554/eLife.42541.028

• Supplementary file 5. Coefficients from the multilevel model analysis in which all EEG components were added simultaneously to predict variability in task performance due to variability in tonic arousal.
DOI: https://doi.org/10.7554/eLife.42541.029

• Transparent reporting form
DOI: https://doi.org/10.7554/eLife.42541.030

**Data availability**

All data have been deposited at https://figshare.com/s/8d6f461834c47180a444, in association with Newman et al (2017). All analysis scripts are publicly available at https://github.com/jochemvan-kempen/2019_pupil_decisionMaking (copy archived at https://github.com/elifesciences-publica-tions/2019_pupil_decisionMaking).

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
