## [Decision Letter]

Thank you for submitting your article "Behavioural and neural signatures of perceptual evidence accumulation are modulated by pupil-linked arousal" for consideration by *eLife*. Your article has been reviewed by two peer reviewers, and the evaluation has been overseen by Eran Eldar as the Reviewing Editor, and Richard Ivry as the Senior Editor. The following individuals involved in review of your submission have agreed to reveal their identity: Tobias H Donner (Reviewer #2); Marieke Jepma (Reviewer #3).

The reviewers have discussed the reviews with one another and the Reviewing Editor has drafted this decision to help you prepare a revised submission.

Summary:

This paper addresses an interesting and timely topic: the effects of arousal (indexed by pupil size) on different stages of information processing during perceptual decision making. Recent studies have already linked variation in pupil size to various components of decision making, but the systematic assessment of the link between pupil size, performance, and a set of EEG signals related to specific information-processing stages in this study is novel.

The authors re-analyze a large amount of data – mainly from one previous study (N=80; Newman et al., 2017), and, for some specific questions, also from other previously published studies. They quantify a variety of EEG markers as a function of pupil-linked arousal: markers of attention, decision processing, and motor preparation, whose properties have previously been characterized extensively.

The relationship between trial-to-trial variations in pupil measures and these EEG markers reveals a non-monotonic relationship of baseline pupil diameter with both reaction time and pre-trial alpha power, which accords with a body of work on baseline arousal and neuromodulatory tone. Interestingly, the authors also find EEG markers (CPP ITCP, N2c, LHB) that scale linearly with baseline pupil diameter, which illuminate the difference between slow performance associated with low and high baseline pupil diameter.

The analyses also yield non-monotonic relationships with task-evoked changes in pupil diameter. This contradicts the monotonic relationships between phasic neuromodulatory or pupil responses and cortical signals or parameters of choice behavior observed in a substantial body of previous work.

We find the topic of this paper important, and we sympathize with the general approach pursued in this study. The data look extremely clean, and the analyses are generally sophisticated. That said, we have some fundamental concerns about the validity of the study's conclusions, in particular those conclusions drawn based on the phasic pupillary responses. In this respect, we think that the conclusions are not sufficiently supported by the data, and the conclusions drawn from these results may, in fact, mislead the community. We thus feel it would be irresponsible to recommend it for publication at *eLife* in its current form. Below, we explain our concerns in detail and make specific suggestions for how to address them. Most importantly, we propose that the authors focus the paper on the baseline pupil diameter findings, which in our estimation are more well founded.

Essential revisions:

1) The increase of pupil response amplitude with RT can be explained by the characteristics of the peripheral system controlling pupil diameter rather than central arousal state – any central arousal signal shaping cortical state (i.e. a response of the LC, basal forebrain, or any other center contributing to central arousal and affecting pupil diameter) passes through a sluggish peripheral system (peripheral nerves/ganglia, smooth pupil muscles) to produce a measured pupil response. Previous work has quantified this transformation as a simple linear convolution with a pupil impulse response function (IRF, e.g., Hoeks and Levelt, 1993; Korn and Bach, JoV, 2016).) This linear systems approach may, or may not, be accurate for the intermediate range of pupil diameters measured in human laboratory settings. Regardless of the specifics of the transformation, the point is that, just as for fMRI, the measured pupil response is only a low-pass filtered version of the central input signal. Obviously this study is about the inferred amplitude of the central arousal signal. If that central signal is sustained, the amplitude of the peripheral pupil response increases with central signal duration – even if the amplitude of the central signal does not change.

Previous work shows that pupil dilation during perceptual decisions is driven by a sustained central input – either a plateau or a slow build-up – throughout the interval of decision formation (Cheadle et al., 2014; de Gee et al., 2014, 2017; Murphy et al., 2016). Whenever there is such a sustained component in the input, it can trivially predict the right (steeper) flank of the plot in Figure 1E – hence, it can also explain part of the subsequent effects of pupil response on EEG markers. This trivial effect may superimpose with an interesting, genuine effect of phasic arousal (perhaps negative slope), to produce the non-monotonic pattern shown in Figure 1E and the subsequent ones. The authors themselves refer to the above conclusion that some component of the central arousal signals may be sustained. (It might not be the LC-component, but DA neurons, some of which exhibit sustained activity during decision formation.) But then the above 'RT-confound' is inescapable. The possibility of such a trivial confound renders the results difficult to interpret.

One possible way to resolve this confound could be to fit different GLMs convolving each input regressor with a "canonical" IRF (Hoeks and Levelt). Within this framework the authors could model various different kinds of input to the pupil system, including both linear and non-linear parametric modulation by RT, and use model comparison to actually test what form the input to the pupil system is actually likely to take place in their data. However, even if this confound is resolved, the present interpretation of the pupil response findings would still be problematic due to its claim to causality (see comment #2).

2) The claim of causality. The paper, as it stands, interprets the findings associated with lower or higher pupillary responses as being caused by differences in phasic arousal. This would be one way to explain the findings. However, there is another way which does not assume causality and seems at least as plausible. Specifically, some long RTs may reflect the subject doing less and other long RTs may reflect the subject doing more. 'Doing more' here could be for instance a more sluggish processing of the stimulus, which leads to lower phasic arousal/effort, and 'doing less' here could be taking extra care to verify detection before responding, and comparing against the stimulus on the other side of the screen (as suggested by Figure 4A), which leads to greater or more sustained phasic arousal/effort. A combination of these two types of trials can parsimoniously explain a U-shape relationship between pupil response and RT while assuming that arousal is simply a result of certain characteristics of the decision process.

It is true that the issue of causality is common to a lot of papers in the field. However, we think this caveat needs to be weighed based on: (1) does the paper strongly argue for a causal interpretation? (2) is there converging evidence supporting the causal perspective (e.g., from both RT and accuracy)? and (3) is there a non-causal interpretation that seems more plausible than the causal one? Our feeling is that all three factors work against the pupil response findings in this paper. Therefore, due to the issues described in the first two comments, we propose that the authors either remove the "phasic arousal" part or relegate it to the supplementary material (the latter option requires that the authors perform the additional analysis proposed in comment 1), and instead focus on the "baseline arousal" part. The findings with respect to baseline pupil diameter do not suffer from the same interpretational issues, at least not to the same degree, and they provide a natural extension to the impact of baseline arousal on previous work on the responses of sensory cortical neurons (McGinley et al., 2015). This is a more specific issue than what the authors are alluding to in the title and Abstract, but it is what it is. In sum, with the above, rather fundamental change, the study might provide insight into the role of (baseline) arousal state in some form of visuo-motor decision processing and selective attention.

3) The authors note that participants performed at ceiling and therefore do not analyze accuracy. Does this mean there were no missed responses/errors at all? Please specify the accuracy data.

4) Throughout the manuscript, the authors infer non-monotonic relationships from the superior fit a quadratic model compared to a linear one. This is not sufficient. Consider the two vectors: x = [1 2 3 4 5] and y = [10 5 1 1 1]. A quadratic function would fit the relationship between x and y better than a linear function even though the relationship is monotonic. This issue plagues many of the analyses, and visual inspection of some of the relevant figures does not convince that there is a significant non-monotonic relationship (e.g., Figure 4E).

5) The details and appropriateness of the hierarchical mediation analysis reported in the subsection “The impact of phasic arousal on task performance is mainly mediated by the consistency in evidence accumulation” do not seem entirely clear. Our main concern here is that the approach may be biased by certain assumptions about the "temporal order of the decision-making process" and the associated "successive stages of neural processing". The point is that motor beta power suppression and CPP seem to behave analogously in all analyses. Yet, it is entered in the hierarchical analysis only after CPP. So, the CPP is identified as having a unique role in mediating arousal effects. Does that conclusion change if the order of entering motor beta and CPP is reversed? We assume that the authors' underlying assumption is as follows: (i) The CPP is generated at a processing stage (e.g. LIP) "prior to" motor beta and (ii) it encodes the accumulating decision variable, which is only subsequently translated into a motor preparatory signal evident in motor beta-band activity. We are convinced that the CPP is a very interesting signal worth studying, but we do not think these assumptions are supported by the available evidence. For example, one of the authors' recent studies (Twomey et al., 2016) has shown that motor beta behaves more similar to the LIP signal than the CPP in a delayed version of the classic Shadlen and Newsome task: encoding upcoming choice throughout the delay. Plus, the authors are aware of all the recent provocative animal work from several different labs (inactivation studies and sophisticated analyses, in rodents and monkeys) that cast doubt even on the notion that LIP/PPC is "the evidence accumulator". We are not saying that this notion is "wrong", but we feel it cannot be treated as an established fact, on which to base the logic of an analysis like the above.

6) Implications for accounts of evidence accumulation are unclear – we do not think that much can be concluded from these results about "evidence accumulation", a term that occurs throughout the title, Abstract, conclusions, etc. Neither do authors present behavioral evidence for such accumulation, nor is it clear why that would be necessary to solve the task at hand: coherence change detection task with strong sensory evidence (50% coherence) that is constant across trials. Detecting such a strong motion signal, and discriminating its coarse direction, is trivial; the only performance-limiting factors are spatial and temporal uncertainty about the occurrence of the change. Indeed, accuracy seems to be "at ceiling" (in Newman et al., it ranged between 92% and 100%). Especially if the change is as large as here, such a task can be solved without any meaningful temporal integration (i.e., over timescales beyond 100 ms, as used for near-threshold motion discrimination). Transient detection, for which there are mechanisms in V4 (Gardner et al., Neuron, 2005; Costagli et al., 2014), would suffice. Please note that the uncertainty about signal onset also renders standard drift diffusion unsuitable for this task (Ossmy et al., 2013; Glaze et al., 2015). Thus, any conclusions about evidence accumulation should be reframed in terms of "decision processing", "motion detection", etc. At the same time, the lack of choice variability across trials severely limits the mechanistic insight that can be gained from this study, because no other behavioral readout than RT is available and fitting models as in the Ossmy et al., 2013 or Glaze et al., 2015 work will not work (simulations form such models, or fits from simpler models, might be possible and informative though).

7) In general, we feel that the current presentation rests on several strong and doubtful assumptions about the internal process underlying behavior, and about the functional nature of the different EEG markers. Some of those assumptions are made explicit, others remain implicit. Here are some paraphrased examples: "CPP equals the decision variable"; "if we detect a CPP in our task, this implies evidence accumulation, because the CCP was linked to evidence accumulation in previous work using other tasks". But here, CPP/motor beta could just be an abrupt step-like signal upon transient motion detection, but variability in the motor component of non-decision time would smooth the associated trial-averaged traces to make it look as if they were gradually building. These assumptions should be made explicit. In fact, we would encourage the authors to try re-frame the paper without these assumptions, most of are not necessary for a presentation of the findings. Thus, Though we appreciate the careful previous work of these labs to characterize these EEG markers, we think it is sufficient to refer to this previous work to motivate the focus on these markers, and then simply report how these markers behave in the current work.

[Editors' note: further revisions were requested prior to acceptance, as described below.]

Thank you for resubmitting your work entitled "Behavioural and neural signatures of perceptual decision-making are modulated by pupil-linked arousal" for further consideration at eLife. Your revised article has been favorably evaluated by Richard Ivry (Senior Editor) and Eran Elder (Reviewing Editor).

The reviewers’ comments were comprehensively addressed, and the manuscript has been substantially improved. There are only minor issues left, brought about by the updated results, that need to be addressed before acceptance.

First, there is remarkable similarity between the baseline diameter and pupil response results. These two measures are correlated, and both are associated with high baseline alpha power and with a similar behavioral effect that is in both cases mediated by the consistency of the CPP. This is all the more notable since the authors made sure to remove variance in the pupil responses that is attributable to differences in baseline diameter. This suggests that both might be in large part two noisy measures of baseline arousal, from which neither measure can be fully dissociated by removing variance attributable to the other measure. Importantly, there were also differences between the measures, but this issue – both the overlap and dissociation – deserves to be pointed out and discussed, especially considering other work indicating that pupil responses are inversely related to baseline LC function.

Second, in the last sentence of the Abstract, confusion can be prevented by pointing out that the terms "attentional engagement, sensory processing and decision processing" relate to EEG correlates of these processes.

---

## [Author Response]

Essential revisions:1) The increase of pupil response amplitude with RT can be explained by the characteristics of the peripheral system controlling pupil diameter rather than central arousal state – Any central arousal signal shaping cortical state (i.e. a response of the LC, basal forebrain, or any other center contributing to central arousal and affecting pupil diameter) passes through a sluggish peripheral system (peripheral nerves/ganglia, smooth pupil muscles) to produce a measured pupil response. Previous work has quantified this transformation as a simple linear convolution with a pupil impulse response function (IRF, e.g., Hoeks and Levelt, 1993; Korn and Bach, JoV, 2016).) This linear systems approach may, or may not, be accurate for the intermediate range of pupil diameters measured in human laboratory settings. Regardless of the specifics of the transformation, the point is that, just as for fMRI, the measured pupil response is only a low-pass filtered version of the central input signal. Obviously this study is about the inferred amplitude of the central arousal signal. If that central signal is sustained, the amplitude of the peripheral pupil response increases with central signal duration – even if the amplitude of the central signal does not change.Previous work shows that pupil dilation during perceptual decisions is driven by a sustained central input – either a plateau or a slow build-up – throughout the interval of decision formation (Cheadle et al., 2014; de Gee et al., 2014, 2017; Murphy et al., 2016). Whenever there is such a sustained component in the input, it can trivially predict the right (steeper) flank of the plot in Figure 1E – hence, it can also explain part of the subsequent effects of pupil response on EEG markers. This trivial effect may superimpose with an interesting, genuine effect of phasic arousal (perhaps negative slope), to produce the non-monotonic pattern shown in Figure 1E and the subsequent ones. The authors themselves refer to the above conclusion that some component of the central arousal signals may be sustained. (It might not be the LC-component, but DA neurons, some of which exhibit sustained activity during decision formation.) But then the above 'RT-confound' is inescapable. The possibility of such a trivial confound renders the results difficult to interpret.One possible way to resolve this confound could be to fit different GLMs convolving each input regressor with a "canonical" IRF (Hoeks and Levelt). Within this framework the authors could model various different kinds of input to the pupil system, including both linear and non-linear parametric modulation by RT, and use model comparison to actually test what form the input to the pupil system is actually likely to take place in their data. However, even if this confound is resolved, the present interpretation of the pupil response findings would still be problematic due to its claim to causality (see comment #2).

We acknowledge that our scalar measure of pupil dilation may indeed have been confounded by the duration of the trial. We took up the reviewers’ advice and tested the fit of various GLMs in order to investigate the form of the input to the pupil system. With this approach, we focused on the different forms previously tested by Murphy et al., 2016. Next, we extended this analysis by applying the best fitting GLM on each individual trial by considering each trial as a separate condition (similar to the approach by Bach et al., 2018. Molecular Psychiatry). This allowed us to estimate the presence of each of the three components (target onset, a sustained period and the response) for each trial. We then used the estimates of the target onset response as scalar values by which we sorted the data in order to investigate their relationship to behavioural performance as well as the EEG measures associated with perceptual decision-making. This analysis has allowed us to reduce the confounding effect of RT on the scalar measure of the pupil response, as well as distinguishing between the target onset and sustained component, whilst still having the single-trial resolution to investigate its relationship to the behavioural and EEG signatures under investigation. With this approach, we found that the relationship between the pupil response to target onset and task performance (and EEG components) are no longer best described by a non-monotonic (U-shaped) relationship. Rather, they are best described by an inverse monotonic relationship, where the largest pupil responses are predictive of the best task performance. The changes to the analysis can now be found in the revised manuscript (Materials and methods), with Figure 1—figure supplements 1-4 relating to this new analysis. Altogether, we believe that this change to the approach of estimating the pupil response deals with the concerns mentioned above, as well as (some of) the concerns raised by the reviewers in comment 2 below.

2) The claim of causality. The paper, as it stands, interprets the findings associated with lower or higher pupillary responses as being caused by differences in phasic arousal. This would be one way to explain the findings. However, there is another way which does not assume causality and seems at least as plausible. Specifically, some long RTs may reflect the subject doing less and other long RTs may reflect the subject doing more. 'Doing more' here could be for instance a more sluggish processing of the stimulus, which leads to lower phasic arousal/effort, and 'doing less' here could be taking extra care to verify detection before responding, and comparing against the stimulus on the other side of the screen (as suggested by Figure 4A), which leads to greater or more sustained phasic arousal/effort. A combination of these two types of trials can parsimoniously explain a U-shape relationship between pupil response and RT while assuming that arousal is simply a result of certain characteristics of the decision process.It is true that the issue of causality is common to a lot of papers in the field. However, we think this caveat needs to be weighed based on: (1) does the paper strongly argue for a causal interpretation? (2) is there converging evidence supporting the causal perspective (e.g., from both RT and accuracy)? and (3) is there a non-causal interpretation that seems more plausible than the causal one? Our feeling is that all three factors work against the pupil response findings in this paper. Therefore, due to the issues described in the first two comments, we propose that the authors either remove the "phasic arousal" part or relegate it to the supplementary material (the latter option requires that the authors perform the additional analysis proposed in comment 1), and instead focus on the "baseline arousal" part. The findings with respect to baseline pupil diameter do not suffer from the same interpretational issues, at least not to the same degree, and they provide a natural extension to the impact of baseline arousal on previous work on the responses of sensory cortical neurons (McGinley et al., 2015). This is a more specific issue than what the authors are alluding to in the title and Abstract, but it is what it is. In sum, with the above, rather fundamental change, the study might provide insight into the role of (baseline) arousal state in some form of visuo-motor decision processing and selective attention.

Indeed, given this dataset, we cannot control for the behavioural approach taken by the subject on individual trials. Differences in approach, however, would likely produce differences between target onset and response, during the sustained period of the task. The target onset component, arguably reflecting a bottom up stimulus processing response (potentially a transient locus coeruleus response), does not suffer from the same interpretational issues. Moreover, these results are in line with previous studies revealing an increase in performance on trials with a larger phasic arousal response, and reveal new insights into how this phasic response influences different neural processing stages of perceptual decision-making. We believe that by focusing on the pupillary response to target onset (separately from the sustained period and response related component), we have addressed the first two of the reviewers’ concerns, and we therefore included these results in the main text, illustrated in Figure 1-3. We, however, welcome any further comments, and if the reviewers still request that we move these results, we are willing to do this.

3) The authors note that participants performed at ceiling and therefore do not analyze accuracy. Does this mean there were no missed responses/errors at all? Please specify the accuracy data.

We specified the accuracy data in the Results, under the header “Both tonic and phasic arousal are predictive of task performance”.

4) Throughout the manuscript, the authors infer non-monotonic relationships from the superior fit a quadratic model compared to a linear one. This is not sufficient. Consider the two vectors: x = [1 2 3 4 5] and y = [10 5 1 1 1]. A quadratic function would fit the relationship between x and y better than a linear function even though the relationship is monotonic. This issue plagues many of the analyses, and visual inspection of some of the relevant figures does not convince that there is a significant non-monotonic relationship (e.g., Figure 4E).

To test for the statistical presence of (inverted) U-shaped relationships in the case of a non-monotonic relationship, we implemented the ‘two-lines’ approach (Simonsohn, 2017). Using this method, we tested whether there is a significant sign change from low to high pupil diameter values. This is done by estimating two linear regression lines separately to low and high pupil diameter values. To classify a relationship as U-shaped, the regression coefficients need to be independently significant and be of opposite sign. The details for this analysis are described in the Materials and methods, subsection “Statistical analyses”.

5) The details and appropriateness of the hierarchical mediation analysis reported in the subsection “The impact of phasic arousal on task performance is mainly mediated by the consistency in evidence accumulation” do not seem entirely clear. Our main concern here is that the approach may be biased by certain assumptions about the "temporal order of the decision-making process" and the associated "successive stages of neural processing". The point is that motor beta power suppression and CPP seem to behave analogously in all analyses. Yet, it is entered in the hierarchical analysis only after CPP. So, the CPP is identified as having a unique role in mediating arousal effects. Does that conclusion change if the order of entering motor beta and CPP is reversed? We assume that the authors' underlying assumption is as follows: (i) The CPP is generated at a processing stage (e.g. LIP) "prior to" motor beta and (ii) it encodes the accumulating decision variable, which is only subsequently translated into a motor preparatory signal evident in motor beta-band activity. We are convinced that the CPP is a very interesting signal worth studying, but we do not think these assumptions are supported by the available evidence. For example, one of the authors' recent studies (Twomey et al., 2016) has shown that motor beta behaves more similar to the LIP signal than the CPP in a delayed version of the classic Shadlen and Newsome task: encoding upcoming choice throughout the delay. Plus, the authors are aware of all the recent provocative animal work from several different labs (inactivation studies and sophisticated analyses, in rodents and monkeys) that cast doubt even on the notion that LIP/PPC is "the evidence accumulator". We are not saying that this notion is "wrong", but we feel it cannot be treated as an established fact, on which to base the logic of an analysis like the above.

The hierarchical mediation analysis allows for the sequential testing of likelihood ratio tests, which could indeed, as the reviewers suggested, be based upon false assumptions about the temporal order of the EEG components in the decision making process. To test whether any components that were excluded from the final model based on this hierarchical approach exerted any significant influence on variability in task performance, we additionally investigated the coefficients of the final regression model (the last step in the hierarchical model analysis) that included all EEG components. This revealed no additional components to those included based on the hierarchical model analysis. These results are summarised in Supplementary files 3 and 5.

6) Implications for accounts of evidence accumulation are unclear – we do not think that much can be concluded from these results about "evidence accumulation", a term that occurs throughout the title, Abstract, conclusions, etc. Neither do authors present behavioral evidence for such accumulation, nor is it clear why that would be necessary to solve the task at hand: coherence change detection task with strong sensory evidence (50% coherence) that is constant across trials. Detecting such a strong motion signal, and discriminating its coarse direction, is trivial; the only performance-limiting factors are spatial and temporal uncertainty about the occurrence of the change. Indeed, accuracy seems to be "at ceiling" (in Newman et al., it ranged between 92% and 100%). Especially if the change is as large as here, such a task can be solved without any meaningful temporal integration (i.e., over timescales beyond 100 ms, as used for near-threshold motion discrimination). Transient detection, for which there are mechanisms in V4 (Gardner et al., Neuron, 2005; Costagli et al., 2014), would suffice. Please note that the uncertainty about signal onset also renders standard drift diffusion unsuitable for this task (Ossmy et al., 2013; Glaze et al., 2015). Thus, any conclusions about evidence accumulation should be reframed in terms of "decision processing", "motion detection", etc. At the same time, the lack of choice variability across trials severely limits the mechanistic insight that can be gained from this study, because no other behavioral readout than RT is available and fitting models as in the Ossmy et al. (2013) or Glaze et al. (2015) work will not work (simulations form such models, or fits from simpler models, might be possible and informative though).

We addressed this concern in this revision by investigating the early target selection signals (N2c) aligned to both target onset as well as the response. If variability in RT was indeed mainly caused by variability in the target onset transient, and there would be a fixed delay between the target onset transient and the response, then the amplitude of the N2c would be much larger when aligned to response compared to target onset. We, however, found that this is not the case. The amplitude of the response-aligned N2c was significantly lower than when aligned to target onset. Insofar as the N2c reflects the target onset transient, this indicates that there is variation in the duration between early target selection and the response. Whether this variation does indeed reflect variability in the duration of evidence accumulation can, however, still be questioned because of the points raised above. Subjects could employ a different strategy on different trials. On some trials they could verify coherent motion in one of the stimuli by comparing it against the other stimulus, whereas on other trials this approach may not have been taken. We address this limitation both in the Results and in the Discussion of the revised manuscript. Additionally, we have substantially reduced our usage of the term evidence accumulation throughout the manuscript.

7) In general, we feel that the current presentation rests on several strong and doubtful assumptions about the internal process underlying behavior, and about the functional nature of the different EEG markers. Some of those assumptions are made explicit, others remain implicit. Here are some paraphrased examples: "CPP equals the decision variable"; "if we detect a CPP in our task, this implies evidence accumulation, because the CCP was linked to evidence accumulation in previous work using other tasks". But here, CPP/motor beta could just be an abrupt step-like signal upon transient motion detection, but variability in the motor component of non-decision time would smooth the associated trial-averaged traces to make it look as if they were gradually building. These assumptions should be made explicit. In fact, we would encourage the authors to try re-frame the paper without these assumptions, most of are not necessary for a presentation of the findings. Thus, Though we appreciate the careful previous work of these labs to characterize these EEG markers, we think it is sufficient to refer to this previous work to motivate the focus on these markers, and then simply report how these markers behave in the current work.

As indicated above (comment 6), we have addressed the limitations of our measure of the decision variable both in the Results and the Discussion. As part of this, we have provided more background, such as making explicit the assumption that the CPP could be a build-to-threshold decision variable, but might equally be a step-wise function that appears like a build-to-threshold decision variable when averaged across trials.

[Editors' note: further revisions were requested prior to acceptance, as described below.]

First, there is remarkable similarity between the baseline diameter and pupil response results. These two measures are correlated, and both are associated with high baseline alpha power and with a similar behavioral effect that is in both cases mediated by the consistency of the CPP. This is all the more notable since the authors made sure to remove variance in the pupil responses that is attributable to differences in baseline diameter. This suggests that both might be in large part two noisy measures of baseline arousal, from which neither measure can be fully dissociated by removing variance attributable to the other measure. Importantly, there were also differences between the measures, but this issue – both the overlap and dissociation – deserves to be pointed out and discussed, especially considering other work indicating that pupil responses are inversely related to baseline LC function.

We addressed the correlation between baseline pupil diameter and the pupil response as well as the overlap in their effect on the investigated EEG signatures of decision-making in the Discussion (section “The overlap and dissociation between baseline pupil diameter and the pupil response”).

Second, in the last sentence of the Abstract, confusion can be prevented by pointing out that the terms "attentional engagement, sensory processing and decision processing" relate to EEG correlates of these processes.

We addressed this issue by explicitly mentioning the term “EEG markers of” attentional engagement, sensory processing and decision processing in the last sentence of the Abstract.